# d2: Improving Reasoning In Diffusion Language Models Via Trajectory Likelihood Estimation

**Guanghan Wang** [1]  **Gilad Turok** [1]  **Yair Schiff** [1]  **Marianne Arriola** [1]  **Volodymyr Kuleshov** [1]

## Abstract

While diffusion language models (DLMs) have achieved competitive performance in text generation, improving their reasoning ability with reinforcement learning remains an active research area. Here, we introduce d2, a reasoning framework tailored for masked DLMs. Central to our framework is a new policy gradient algorithm that relies on accurate estimates of the sampling trajectory likelihoods. Because computing these likelihoods naively is computationally expensive for masked DLMs, we develop a family of estimators tailored to distinct model classes. For DLMs that support a sampling algorithm called any-order decoding, we propose d2-AnyOrder, which achieves exact trajectory likelihood with a single model pass. Through an empirical study of widely used DLMs, we show that any-order decoding is not universally supported in practice. For standard masked diffusion models, we propose d2-StepMerge, which approximates the trajectory likelihood, trading off compute for approximation accuracy in an analytically tractable manner. Empirically, d2 significantly outperforms widely-used RL baselines when applied to popular DLMs, and sets a new state-of-the-art performance for DLMs on logical reasoning tasks (Countdown and Sudoku) and math reasoning benchmarks (GSM8K and MATH500). We provide the code along with a blog post on the project page: https://guanghanwang.com/d2

## 1. Introduction

Diffusion language models (DLMs) (Nie et al., 2025; Ye et al., 2025; Gong et al., 2026; Song et al., 2025) have recently emerged as a competitive alternative to autoregressive (AR) models for text generation, featuring attractive properties such as controllability (Nisonoff et al., 2025; Schiff et al., 2025) and fast parallel generation (Wang et al., 2025; Khanna et al., 2025). Yet, while reinforcement learning (RL) has become the de-facto approach for inducing reasoning in AR LLMs, post-training DLMs using RL remains an active research area.

Here, we introduce d2, a reasoning framework that improves over previous approaches in the setting of masked DLM, a popular type of diffusion language models. Central to our framework is a new policy gradient algorithm that requires accurate estimation of the likelihood of sampling trajectories. Because computing these likelihoods naively is computationally expensive for masked DLMs, we develop a family of estimators tailored to distinct model classes.

Concretely, this family consists of two estimators. The first, d2-AnyOrder, computes likelihoods exactly in a single forward pass for DLMs that support a sampling algorithm called any-order decoding (AO-dLLMs); through an empirical study, we show that popular masked diffusion models such as LLaDA (Nie et al., 2025) do not satisfy this property by default. The second, d2-StepMerge, targets standard masked diffusion models (MDMs) and evaluates trajectories only at specific steps, reducing forward passes while controlling the approximation error; we theoretically study this error via a bound that decreases with the number of steps, quantifying a compute-bias tradeoff.

Empirically, d2 improves reasoning without supervised chain-of-thought finetuning (Wei et al., 2022): when applied to DLMs that support any-order decoding, d2 significantly outperforms widely-used RL baselines. When applied to LLaDA-8B-Instruct, it surpasses prior diffusion-based RL methods on logical (Countdown, Sudoku) and mathematical (GSM8K, MATH500) benchmarks, establishing a new state of the art for DLMs under matched FLOPs.

In summary, our work makes the following contributions: (1) We introduce d2, a reasoning framework for RL post-training of masked DLMs, consisting of a novel GRPO-style policy gradient and a family of trajectory likelihood estimators—highlighting the central role of accurate likelihood estimation for training reasoning DLMs. (2) For

---

[1]Cornell University, Cornell Tech. Correspondence to: Guanghan Wang <gw354@cornell.edu>.

*Proceedings of the 43$^{rd}$ International Conference on Machine Learning*, Seoul, South Korea. PMLR 306, 2026. Copyright 2026 by the author(s).

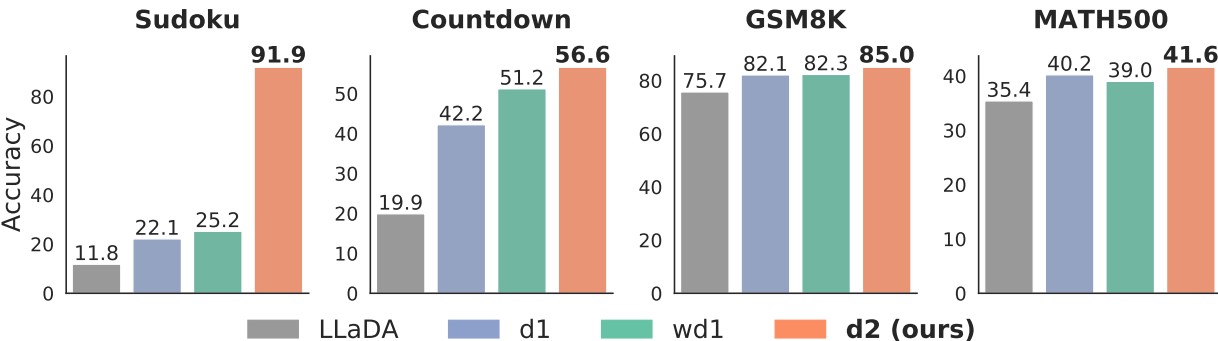

*Figure 1.* Benchmark performance of different RL post-training algorithms applied to LLaDA-8B-Instruct (Nie et al., 2025). Without supervised finetuning (SFT), d2 outperforms d1 (Zhao et al., 2025) with SFT and wd1 (Tang et al., 2026) on all four reasoning benchmarks.

DLMs that support any-order decoding (AO-dLLMs), we propose d2-AnyOrder, which provides unbiased one-pass likelihood estimates. Empirically, d2-AnyOrder significantly outperforms widely-used RL baselines such as diffu-GRPO (Zhao et al., 2025) and DDPO (Black et al., 2024). (3) For standard MDMs that do not natively support any-order decoding, we propose d2-StepMerge, a deterministic likelihood estimator with an analytically bounded approximation error. When applied to LLaDA-8B-Instruct, d2-StepMerge achieves state-of-the-art reasoning results on Sudoku, Countdown, GSM8K, and MATH500.

## 2. Background

### 2.1. Masked Diffusion Language Models

Diffusion language models (DLMs) are characterized by two processes. The first is a pre-defined corruption (also know as forward) process $q$. This process adds noise to a 'clean' token $\mathbf{x}$ drawn from the data distribution to produce progressively noisy latents $\mathbf{x}_t (t \in [0, 1])$, with noise levels increasing in $t$, and terminates at the fully corrupted latent $\mathbf{x}_1$ drawn from a simple prior. The second process is the learned denoising (backward) process $p_\theta$, which is trained to undo the corruptions from the forward trajectory.

Recent discrete diffusion models have focused on forward processes that interpolate between signal and noise (Austin et al., 2021a), and in particular they rely on a specific corruption process known as masked diffusion (referred to as MDMs henceforth; Sahoo et al. (2024); Shi et al. (2024); Ou et al. (2025)). The marginals of this process reflect probability mass iteratively moving away from the data and towards a special mask token $\mathbf{m}$: $q(\mathbf{x}_t \mid \mathbf{x}) = \alpha_t \mathbf{x} + (1 - \alpha_t)\mathbf{m}$, where $\alpha_t$ is a noise schedule monotonically decreasing in $t$. Given a data token sequence $\mathbf{x}_0^{1:L}$, the backward process is learned by optimizing the following variational lower bound:

$$\mathcal{L}_{\text{MDLM}} = \mathbb{E}_{\mathbf{x}_t^{1:L} \sim q(\cdot|\mathbf{x}_0^{1:L})} \left[ \frac{\alpha_t'}{1 - \alpha_t} \sum_{l=1}^{L} \log p_\theta(\mathbf{x}_0^l \mid \mathbf{x}_t^{1:L}) \right].$$
(1)

### 2.2. Any-order Autoregressive Models

Hoogeboom et al. (2022) and Ou et al. (2025) demonstrate that the MDLM training objective can be transferred into an any-order autoregressive model (AO-ARM) variant, thus enabling the model to generate sequences autoregressively but not necessarily in a left-to-right dependency. In particular, denoting $\sigma$ as a permutation of integers $1, \ldots, L$, and $S(D)$ as the set of all possible permutations, we have:

$$\mathcal{L}_{\text{MDLM}} = \mathcal{L}_{\text{AO-ARM}} =$$
$$\mathbb{E}_{\sigma \sim U(S_D)} \left[ \sum_{l=1}^{L} \log p_\theta(\mathbf{x}_0^{\sigma(l)} \mid \mathbf{x}_0^{\sigma(<l)}) \right].$$
(2)

### 2.3. Reinforcement Learning with Verifiable Rewards

Policy gradient methods (Williams, 1992; Sutton et al., 1999) have become a central paradigm for improving the reasoning ability of large language models during post-training (Ouyang et al., 2022; Ahmadian et al., 2024; Bai et al., 2022; Li et al., 2024). Starting from a pre-trained model $\pi_{\text{ref}}$, reinforcement learning-based reasoning algorithms optimize a new policy network $\pi_\theta$ by ascending the gradient of the expected reward $r$ for completions generated from $\pi_\theta$ conditioned on an input question $\mathbf{q}$:

$$\nabla_\theta \mathbb{E}_{\mathbf{x}^{1:L} \sim \pi_\theta(\cdot|\mathbf{q})} \left[ r(\mathbf{x}^{1:L}, \mathbf{q}) \right],$$
(3)

where $\mathbf{x}^{1:L}$ denotes a model-generated answer consisting of $L$ tokens. A widely used approach in this context is Group Relative Policy Optimization (GRPO; Shao et al. (2024)), which provides a computationally efficient and low-variance estimator for policy updates. For each query $\mathbf{q}$, GRPO samples a group of $G$ candidate answers $\{\mathbf{x}_{(1)}^{1:L_{(1)}}, \mathbf{x}_{(2)}^{1:L_{(2)}}, \ldots, \mathbf{x}_{(G)}^{1:L_{(G)}}\}$ from a stale policy $\pi_{\text{old}}$. Each answer is then assigned an advantage value

$A_{(i)} := \frac{r(\mathbf{x}_{(i)}^{1:L(i)}, \mathbf{q}) - \text{mean}\{r(\mathbf{x}_{(j)}^{1:L(j)})\}_{1 \le j \le G}}{\text{std}\{r(\mathbf{x}_{(j)}^{1:L(j)})\}_{1 \le j \le G}}$. GRPO employs the following clipped objective to optimize AR models:

$$-\mathbb{E}_{\mathbf{x}^{1:L} \sim \pi_{\text{old}}(\cdot|\mathbf{q})} \left[ \frac{1}{L} \sum_{l=1}^{L} \min(\rho^l A^l, \text{clip}(\rho^l, 1-\epsilon, 1+\epsilon)A^l) \right.$$
$$\left. + \beta D_{\text{KL}}\Big( \pi_\theta(\mathbf{x}^{1:L} \mid \mathbf{q}) \,\|\, \pi_{\text{ref}}(\mathbf{x}^{1:L} \mid \mathbf{q}) \Big) \right],$$
$$(4)$$

where $\rho^l := \frac{\pi_\theta(\mathbf{x}^l|\mathbf{x}^{<l},\mathbf{q})}{\pi_{\text{old}}(\mathbf{x}^l|\mathbf{x}^{<l},\mathbf{q})}$ denotes the per-token importance ratio, and the same advantage value is assigned to each token in a sequence. The clipping parameter $\epsilon$ constrains policy updates within a trust region, and the KL regularization term penalizes divergence from the reference policy $\pi_{\text{ref}}$. Importantly, for standard AR models, due to the causal structure of their attention masks, the summation over $L$ tokens can be efficiently computed in a single model pass.

## 3. Method

### 3.1. Reinforcement Learning Objective for DLMs

In contrast to autoregressive (AR) models, whose likelihood accurately factorizes across token positions, the exact likelihood of diffusion language models (DLMs) is computationally intractable. This structural difference renders it theoretically unjustified to directly apply the AR policy gradient formula to DLMs, and makes the derivation of a policy gradient for DLMs a non-trivial problem.

In this section, we introduce our derivation from the policy gradient formula to a modern GRPO RL objective for masked DLMs. To begin, we define masked DLMs' policy gradient objective with respect to the final denoised tokens $\mathbf{x}_0^{1:L}$:

$$\nabla_\theta \mathcal{J}(\theta) = \nabla_\theta \mathbb{E}_{\mathbf{x}_0^{1:L} \sim \pi_\theta(\cdot|\mathbf{q})} \left[ r(\mathbf{x}_0^{1:L}, \mathbf{q}) \right]. \quad (5)$$

Inspired by Black et al. (2024), we marginalize the likelihood over time latents. Moreover, we introduce importance sampling (Kakade & Langford, 2002) to allow reusing of trajectories generated by a stale policy $\pi_{\text{old}}$, which is widely adopted given the computational cost of on-policy sampling. Based on these tricks, we state the following theorem to further simplify $\nabla_\theta \mathcal{J}(\theta)$ (see detailed proof in Appendix 3.1).

**Theorem 3.1.** *At $\theta = \theta_{old}$, $\nabla_\theta \mathcal{J}(\theta)$ admits the following decomposition over latent diffusion steps:*

$$\nabla_\theta \, \mathbb{E}_{\mathbf{x}_{0:T}^{1:L} \sim \pi_{\text{old}}(\cdot|\mathbf{q})} \left[ r(\mathbf{x}_0^{1:L}, \mathbf{q}) \sum_{t=0}^{T-1} \sum_{l=1}^{L} \mathbf{1}_{t,l} \cdot \rho_t^l \right], \quad (6)$$

*where* $\mathbf{1}_{t,l} := \mathbf{1}_{\{\mathbf{x}_{t+1}^l = m, \mathbf{x}_t^l \neq m\}}$*, and* $\rho_t^l := \frac{\pi_\theta(\mathbf{x}_t^l|\mathbf{x}_{t+1}^{1:L}, \mathbf{q})}{\pi_{\text{old}}(\mathbf{x}_t^l|\mathbf{x}_{t+1}^{1:L}, \mathbf{q})}$*.*

*Remark* 3.2. In practice, sequences are sampled once from $\pi_{\text{old}}$ and reused for multiple gradient updates. After the first gradient update, the equivalence in Theorem 3.1 is no longer valid and is just an approximation of the true policy gradient. However, this approximation is justified as long as $\theta$ remains close to $\theta_{\text{old}}$, which is typically enforced by restricting the size of each policy update.

To stabilize training, we adapt Eq. (6) by replacing rewards with advantages (Williams, 1992; Sutton et al., 1999), introducing a clipped trust region (Schulman et al., 2015), and adding a regularization term penalizing divergence from a reference policy (Schulman et al., 2017). Similar to how GRPO averages over sequence lengths, we add a $\frac{1}{L}$ regularization term to cancel out the impact of different sequence lengths within a group. Overall, these operations lead to the GRPO objective for masked diffusion language models.

**Corollary 3.3.** *The GRPO objective for masked DLM is given by*

$$-\mathbb{E}_{\mathbf{x}_0^{1:L} \sim \pi_{\text{old}}} \left[ \sum_{t=0}^{T-1} \frac{1}{L} \sum_{l=1}^{L} \mathbf{1}_{t,l} \min(\rho_t^l A^l, \text{clip}(\rho_t^l, 1-\epsilon, 1+\epsilon)A^l) \right.$$
$$\left. + \beta D_{\text{KL}}\Big( \pi_\theta(\mathbf{x}_{0:T}^{1:L} \mid \mathbf{q}) \,\|\, \pi_{\text{ref}}(\mathbf{x}_{0:T}^{1:L} \mid \mathbf{q}) \Big) \right].$$
$$(7)$$

### 3.2. Estimating the Policy Gradient for DLMs

Evaluating and optimizing the GRPO objective requires computing the likelihoods $\pi_\theta(\mathbf{x}_{0:T}^{1:L}), \pi_{\text{old}}(\mathbf{x}_{0:T}^{1:L}), \pi_{\text{ref}}(\mathbf{x}_{0:T}^{1:L})$ of samples $\mathbf{x}_{0:T}^{1:L}$ taken from the policy. While autoregressive models support computing these likelihoods in a single forward pass, a naive approach in a diffusion model requires $T$ forward passes, which is computationally prohibitive.

Here, we introduce two computationally efficient estimators, each tailored to a distinct class of masked DLMs (Table 1). For DLMs that support a sampling algorithm called **any-order decoding** (AO-dLLMs), we propose the **d2-AnyOrder** estimator, which computes the trajectory likelihood exactly in a single forward pass. For standard masked diffusion models (MDMs) that do not natively support any-order decoding, we propose the **d2-StepMerge** estimator, which approximates the trajectory likelihood using multiple forward passes with an analytically bounded approximation error (Theorem 4.1). We empirically demonstrate in Section 5.2 that d2-StepMerge achieves a superior trade-off between performance and total compute budget compared to existing baselines.

*Table 1.* The two algorithms in d2 and their target model classes.

| Algorithm | Model class | Estimator |
|---|---|---|
| d2-AnyOrder | AO-dLLMs | Exact (one pass) |
| d2-StepMerge | Standard MDMs | Approximate (multiple passes) |

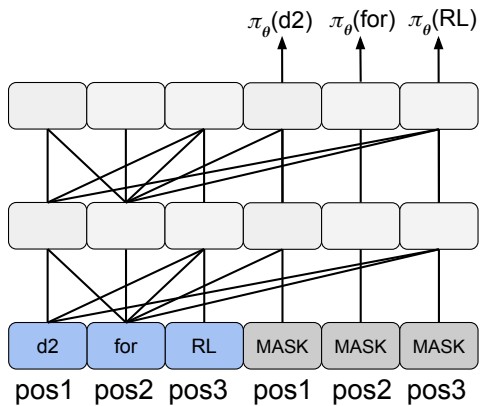

*Figure 2.* Illustration of one-shot trajectory likelihood evaluation. We depict attention with query tokens (one layer up) attending to keys / values (one layer below) via an undirected connected line. The output at each position is depicted with a directed arrow. "pos" refers to positional encoding index. We use a three token example where the decoding order is "for→d2→RL".

### 3.2.1. D2-ANYORDER ESTIMATOR

We now introduce d2-AnyOrder, which computes exact trajectory likelihoods in a single forward pass. This estimator is compatible with transformer-based masked DLMs from which we sample using a simple algorithm called any-order decoding.

Recall that our goal is to estimate the likelihoods $\pi(\mathbf{x}_{0:T}^{1:L})$ of trajectories $\mathbf{x}_{0:T}^{1:L}$ sampled from the policy. Using connections between masked and any-order models, we can express the likelihood of a trajectory as $\pi(\mathbf{x}_{0:T}^{1:L}) = \prod_{l=1}^{L} \pi(\mathbf{x}_0^{\sigma(l)} \mid \mathbf{x}_0^{\sigma(<l)})$. Here, instead of a randomly sampled permutation, $\sigma$ denotes the decoding order of the sample tokens. In other words, $\mathbf{x}_0^{\sigma(<l)}$ denotes the tokens decoded before $\mathbf{x}_0^l$. Our goal is then to compute each conditional likelihood term $\pi(\mathbf{x}_0^{\sigma(l)} \mid \mathbf{x}_0^{\sigma(<l)})$ in one forward pass.

**d2-AnyOrder.** To efficiently compute the $L$ marginals, we propose a technique for packing them into a single sequence of length $2L$ that will be processed in parallel by a transformer. Specifically, we construct a $2L$-length sequence $\mathbf{x}_0^{1:L} \oplus \mathbf{m}^{L+1:2L}$, where $\oplus$ denotes concatenation along the sequence dimension and $\mathbf{m}^{L+1:2L}$ are mask tokens. The positional encodings are assigned as $\text{pos}_l = l \bmod L$ for $1 \leq l \leq 2L$, so that each token–mask pair shares the same position index. We then define the attention mask so that a clean token $\mathbf{x}_0^{\sigma(l)}$ attends to $\mathbf{x}_0^{\sigma(\leq l)}$; mask tokens $\mathbf{m}^{L+\sigma(l)}$ attend to $\mathbf{x}_0^{\sigma(<l)} \cup \mathbf{m}^{L+\sigma(l)}$. We denote the resulting likelihood estimate as $\pi^{\text{AO}}(\mathbf{x}_0^l \mid \mathbf{x}_0^{1:L} \oplus \mathbf{m}^{L+1:2L})$. More details are provided in Figure 2 and Appendix B.3.

When using this estimator, we train the policy network with

the following loss function:

$$
-\mathbb{E}_{\mathbf{x}_{0:T}^{1:L} \sim \pi_{\text{old}}(\cdot \mid \mathbf{q})} \left[ \frac{1}{L} \sum_{l=1}^{L} \min\left( \rho_{n,l}^{\text{AO}} A^l, \text{clip}(\rho_{n,l}^{\text{AO}}, 1-\epsilon, 1+\epsilon) A^l \right) \right.
$$
$$
\left. + \beta D_{\text{KL}}\left( \pi_\theta(\mathbf{x}_{0:T}^{1:L} \mid \mathbf{q}) \,\|\, \pi_{\text{ref}}(\mathbf{x}_{0:T}^{1:L} \mid \mathbf{q}) \right) \right],
$$
(8)

where $\rho_{n,l}^{\text{AO}} := \frac{\pi_\theta^{\text{AO}}(\mathbf{x}_0^l \mid \mathbf{x}_0^{1:L} \oplus \mathbf{m}^{L+1:2L}, \mathbf{q})}{\pi_{\text{old}}^{\text{AO}}(\mathbf{x}_0^l \mid \mathbf{x}_0^{1:L} \oplus \mathbf{m}^{L+1:2L}, \mathbf{q})}$. For simplicity, we also refer to the RL algorithm built on the d2-AnyOrder estimator as d2-AnyOrder.

**When Does The Any-Order Estimator Work?** The above procedure works if the estimated likelihood of each token $\pi^{\text{AO}}(\mathbf{x}_0^l \mid \mathbf{x}_0^{1:L} \oplus \mathbf{m}^{L+1:2L})$ equals the probability $\pi(\mathbf{x}_0^{\sigma(l)} \mid \mathbf{x}_0^{\sigma(<l)})$ of that token during sampling.

---

**Algorithm 1** Any-Order Decoding

**Input:** DLM model $p_\theta$, sequence length $L$.
$\mathbf{x}^{1:L} \leftarrow \mathbf{m}^{1:L}; \sigma(1), \dots, \sigma(L) \leftarrow L+1; n \leftarrow 0$
**while** $n < L$ **do**
  **for** $l = 1$ to $L$ **do**
    $\text{attn}(\mathbf{x}^{\sigma(l)}) \leftarrow \mathbf{x}_0^{\sigma(\leq l)}$ if $\mathbf{x}^{\sigma(l)} \neq \mathbf{m}$ else $\mathbf{x}_0^{\sigma(<l)} \cup \mathbf{m}^{\sigma(l)}$
  **end for**
  $\mathbf{x}_0^{l_{1:k}} \sim p_\theta(\cdot \mid \mathbf{x}^{1:L}, \text{attn})$
  $\mathbf{x}^{l_{1:k}} \leftarrow \mathbf{x}_0^{l_{1:k}}; \sigma(l_j)_{j=1}^k \leftarrow n+j; n \leftarrow n+k$
**end while**
**return** $\mathbf{x}^{1:L}$

---

This property holds by construction when we sample from a masked DLM using the simple algorithm illustrated in Figure 3, which we call any-order decoding (pseudocode provided in Algorithm 1). At each time step, the algorithm inputs a partially masked sequence $\mathbf{x}^{1:L}$. Notably, we remove the time steps $t$ and instead use the notation $\mathbf{x}^{1:L}$. For each position $l$, $\mathbf{x}^l$ can either be a clean token, i.e., $\mathbf{x}_0^l$ or a mask token, i.e., $\mathbf{m}^l$. At each decoding step, we input $\mathbf{x}^{1:L}$ into a transformer to compute logits at every masked token. Then $k$ token positions are selected for unmasking based on some heuristics (e.g., top-$k$ confidence (Nie et al., 2025) or confidence-threshold (Wu et al., 2026)). Unmasked tokens at selected positions are sampled and added to the sequence. Crucially, we set the attention mask of the transformer parameterizing the DLM to satisfy two properties:

- **Independent Masks:** Mask tokens do not attend to each other: they attend only to unmasked tokens and to themselves.

- **Order Causality:** Unmasked tokens attend only to tokens decoded at earlier time steps and to themselves.

**When Does The Any-Order Estimator Not Work?** Any-order decoding can be applied to any masked DLM, which always yields samples whose likelihood can afterwards be computed in a single forward pass. Unfortunately, any-order decoding does not always produce high-quality samples.

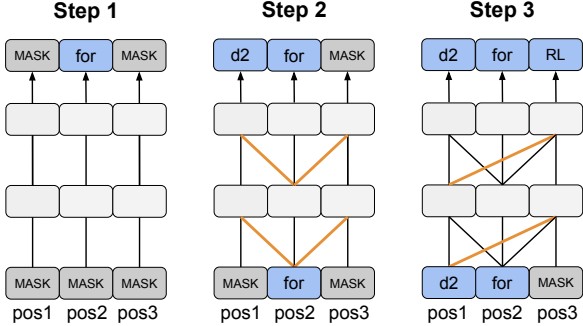

*(a)* Any-order decoding.

*Figure 3.* Illustration of the any-order decoding algorithm for masked DLMs. This example follows the setting of Figure 2 where three tokens are decoded in the order of "for→d2→RL". At each time step, newly added attention relations in any-order decoding are highlighted with red line markers.

*Table 2.* Average ground-truth per token log-likelihood v.s. the d2-AnyOrder estimates applied to LLaDA-8B-Instruct. We also report the KL divergence between the two likelihood distributions.

|  | Per token LL. | $D_{\mathrm{KL}}(\pi_{\mathrm{ref}}||\pi_{\mathrm{ao}})$ |
|---|---|---|
| ground-truth | -0.128 | — |
| d2-AnyOrder | -3.051 | 2.334 |

If the model was not trained with independent masks and order causality, it may not produce good samples when these properties are introduced at inference time.

To further validate this argument, we design an experiment in which we first let LLaDA-8B-Instruct generate sequences from prompts in the GSM8K test set, and then apply d2-AnyOrder to compute their trajectory likelihood. We then compare d2-AnyOrder likelihood with the ground-truth trajectory log-likelihood of LLaDA-8B-Instruct. As shown in Table 2, when directly applied to LLaDA-8B-Instruct, d2-AnyOrder deviates from the ground truth by an order of magnitude. This discrepancy confirms that standard MDLMs do not inherently support any-order decoding. Building on similar observations, recent studies (Sahoo et al., 2026; Arriola & Kuleshov, 2026) have proposed specialized training paradigms for models that natively support any-order decoding, which we utilize to evaluate d2-AnyOrder in Section 5.1.

### 3.2.2. D2-STEPMERGE ESTIMATOR

As noted above, not all DLMs support any-order decoding. For standard masked DLMs, we propose d2-StepMerge, which approximates the trajectory likelihood with multiple forward passes.

Here, we reuse the MDLM decomposition of the trajectory likelihood: $\pi(\mathbf{x}_{0:T}^{1:L}) = \prod_{t=0}^{T-1}\prod_{l=1}^{L}\mathbf{1}_{t,l}\cdot\pi(\mathbf{x}_t^l\mid\mathbf{x}_{t+1}^{1:L})$.

Computing this trajectory likelihood naively takes $T$ model passes, rendering it computationally prohibitive. To address this, we draw inspiration from block composite likelihood (Besag, 1975; Lindsay, 1988; Varin et al., 2011), which approximates intractable joint distributions by products of tractable factors over blocks. Concretely, we cut the sample trajectory of $T$ tokens evenly into $N$ contiguous time segments. For each time segment, we use the output of one model pass as a proxy for token likelihoods within this segment. Formally, the trajectory likelihood is approximated as $\pi(\mathbf{x}_{0:T}^{1:L}) \approx \prod_{n=0}^{N-1}\prod_{l=1}^{L}\mathbf{1}_{n,l}\cdot\pi(\mathbf{x}_{\frac{nT}{N}}^l\mid\mathbf{x}_{\frac{(n+1)T}{N}}^{1:L})$, where $\mathbf{1}_{n,l} := \mathbf{1}_{\{\mathbf{x}_{\frac{(n+1)T}{N}}^l=\mathrm{m},\mathbf{x}_{\frac{nT}{N}}^l\neq\mathrm{m}\}}$.

Based on this estimator, we train the policy network with the following GRPO objective:

$$-\mathbb{E}_{\pi_{\mathrm{old}}}\left[\sum_{n=0}^{N-1}\frac{1}{L}\sum_{l=1}^{L}\mathbf{1}_{n,l}\min\left(\rho_n^l A^l, \mathrm{clip}(\rho_n^l, 1\text{-}\epsilon, 1\text{+}\epsilon)A^l\right)\right.$$
$$\left. + \beta D_{\mathrm{KL}}\left(\pi_\theta(\mathbf{x}_{0:T}^{1:L}\mid\mathbf{q})\,\|\,\pi_{\mathrm{ref}}(\mathbf{x}_{0:T}^{1:L}\mid\mathbf{q})\right)\right], \qquad (9)$$

where $\rho_n^l := \frac{\pi_\theta(\mathbf{x}_{\frac{nT}{N}}^l|\mathbf{x}_{\frac{(n+1)T}{N}}^{1:L},\mathbf{q})}{\pi_{\mathrm{old}}(\mathbf{x}_{\frac{nT}{N}}^l|\mathbf{x}_{\frac{(n+1)T}{N}}^{1:L},\mathbf{q})}$. For simplicity, we also refer to the RL algorithm built on the d2-StepMerge estimator as d2-StepMerge.

**Understanding the role of $N$.** With d2-StepMerge, we reduce the number of model passes from $T$ to $N$. This raises a natural question: "what is the effect of $N$ on the accuracy of the policy gradient?" Since the importance weight is the quotient of two trajectory likelihood evaluations, we further narrow this down to studying the discrepancy between the complete likelihood decomposition of the trajectory and that yielded by our StepMerge strategy. Formally, we quantify the discrepancy $D_N$ between the complete and StepMerge decompositions as:

$$D_N := D_{\mathrm{KL}}\left(\prod_{t=0}^{T-1}\pi_\theta(\mathbf{x}_t|\mathbf{x}_{t+1})\,\|\,\prod_{n=0}^{N-1}\pi_\theta(\mathbf{x}_{\frac{nT}{N}}|\mathbf{x}_{\frac{(n+1)T}{N}})\right).$$

We compute $D_N$ for LLaDA-8B-Instruct (Nie et al., 2025) on the test sets of four datasets (introduced in Section 5.2). As shown in Figure 5, $D_N$ decreases monotonically with increasing $N$, indicating that d2-StepMerge trades off compute for likelihood estimation precision.

*Remark* 3.4. diffu-GRPO (Zhao et al., 2025) is a special case of d2-StepMerge where $N = 1$. As noted above, $N = 1$ produces an inaccurate likelihood estimation and may thus harm the performance of RL. This is consistent with our experimental results shown in Section 5.2.

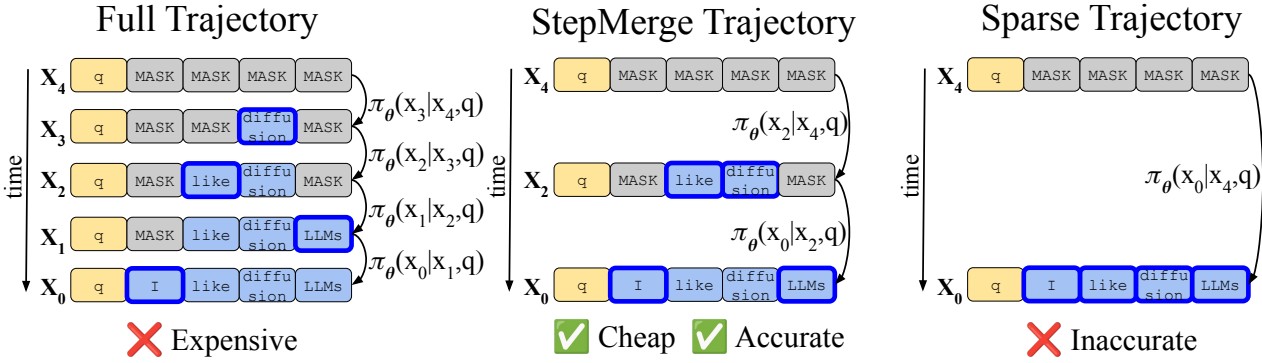

**Figure 4.** Illustration of our proposed d2-StepMerge likelihood estimator. In d2-StepMerge, we cut the trajectory evenly into $N$ time segments and evaluate the likelihood for each segment together. Newly decoded tokens on which we compute the likelihood at the corresponding model forward pass are highlighted.

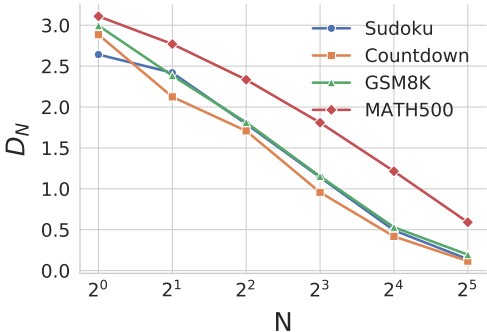

**Figure 5.** $D_N$ of LLaDA-8B-Instruct for varying $N$.

## 4. Theoretical Analysis

In order to quantify the compute-bias trade-off of d2-StepMerge, in this section, we introduce an upper bound for $D_N$ (defined in Section 3.2.2). Our bound monotonically decreases as $N$ increases, justifying our observation in Figure 5.

**Theorem 4.1** (Approximation Error Bound). *Suppose $\pi_\theta$ has a fixed schedule where $L$ tokens are unmasked over $T$ timesteps. The KL divergence $D_N$ between the full trajectory and StepMerge approximation is bounded by:*

$$D_N \le L \cdot \log\left(\frac{T}{N} + 1\right) + L \cdot \epsilon_{block}. \qquad (10)$$

*where $\epsilon_{block}$ is a constant measuring how much $\pi_\theta$'s token predictions can change when we skip intermediate diffusion steps within a block.*

## 5. Experiments

### 5.1. d2-AnyOrder

#### 5.1.1. ESO-LM

**Experimental Setup.** We first evaluate d2-AnyOrder on Eso-LM (Sahoo et al., 2026), a class of DLMs that naturally support any-order decoding. We employ a 190M-parameter model pre-trained on OpenWebText (Gokaslan et al., 2019).

Following Singhal et al. (2025), we assess the capacity of d2-AnyOrder to steer the model toward generating toxic content—a rare behavior in the base model and a critical benchmark for red-teaming scenarios. Toxicity is measured using a pre-trained classifier (Logacheva et al., 2022) as the reward model, where the objective is to maximize the toxicity score.

To conduct a fair comparison, we record the compute budget of different methods (measured in Floating-Point Operations, i.e., FLOPs) and evaluate them at every FLOP interval. For each RL-finetuned checkpoint, we generate 512 sequences and report the average toxicity score generated by the reward model.

**Table 3.** Toxicity Score vs. FLOPs. Our d2-AnyOrder GRPO approach significantly dominates the DDPO baseline in toxicity steering for a given compute budget.

| Method | FLOPs $\times 10^{17}$ | | | | | |
|---|---|---|---|---|---|---|
| | 0.00 | 0.25 | 0.50 | 0.75 | 1.00 | 1.25 |
| DDPO toxicity | -9.2 | -9.2 | -9.1 | -8.9 | -8.9 | -8.6 |
| d2 toxicity (ours) | -9.2 | **-8.5** | **-7.3** | **-5.5** | **-2.7** | **-0.7** |

**Baselines.** Since this setting involves prompt-free, open-ended generation, the likelihood evaluation in diffu-GRPO is uninformative. Consequently, following Su et al. (2026), we adopt DDPO (Black et al., 2024)—a widely used policy gradient RL algorithm for diffusion models—as our primary baseline. Implementation details for the baseline are provided in Appendix C.1.1.

**Results.** As illustrated in Table 3, d2-AnyOrder substantially outperforms DDPO under equivalent compute budgets. Our method achieves near-maximum toxicity scores, i.e., 0, whereas DDPO remains below -8.

**Ablation: d2-AnyOrder v.s. d2-StepMerge.** We evaluate the impact of likelihood estimation exactness on RL by comparing d2-AnyOrder with its approximate counterpart, d2-StepMerge. For d2-StepMerge, we apply $N = 8$ due to its superior empirical performance. As shown in Table 4, d2-AnyOrder consistently outperforms d2-StepMerge across

*Table 4.* Comparison of the toxicity score dynamics between d2-AnyOrder and d2-StepMerge.

| Method | FLOPs $\times 10^{17}$ | | | | | |
|---|---|---|---|---|---|---|
| | 0.00 | 0.25 | 0.50 | 0.75 | 1.00 | 1.25 |
| d2-StepMerge | -9.2 | -8.7 | -8.0 | -6.6 | -4.3 | -1.5 |
| d2-AnyOrder | -9.2 | **-8.5** | **-7.3** | **-5.5** | **-2.7** | **-0.7** |

all compute budgets. This empirical result confirms that the exact trajectory likelihoods enabled by d2-AnyOrder provide a higher-fidelity signal for RL optimization compared to segment-based approximations.

### 5.1.2. SET DIFFUSION

**Experimental Setup.** We next evaluate d2-AnyOrder on a model from a different architecture family. We use the Qwen3-1.7B checkpoint from Arriola & Kuleshov (2026), which finetunes Qwen3 as an any-order dLLM via an AO-ARM objective they propose. As reported in Arriola & Kuleshov (2026), AO finetuning yields substantially better downstream accuracy than Block diffusion finetuning on the same base model (Table 5, top), motivating our focus on the AO variant for RL post-training. We compare d2-AnyOrder against diffu-GRPO under matched compute on GSM8K.

*Table 5.* GSM8K accuracy with Qwen3-1.7B as the base model. SFT results (top) are reported by Arriola & Kuleshov (2026); RL results (bottom, applied on top of AO SFT) are ours. Best result is **bolded**.

| Method | GSM8K |
|---|---|
| Block diffusion SFT (Arriola & Kuleshov, 2026) | 48% |
| AO SFT (Arriola & Kuleshov, 2026) | 62% |
| AO SFT + diffu-GRPO | 63% |
| AO SFT + d2-AnyOrder | **67%** |

**Results.** Under matched compute, d2-AnyOrder reaches 67% on GSM8K, surpassing diffu-GRPO at 63%. This validates that the benefits of d2-AnyOrder extend to an additional any-order dLLM architecture beyond Eso-LM.

### 5.1.3. ANY-ORDER CAUSAL LLADA

**Experimental Setup.** We finally evaluate d2-AnyOrder on a larger any-order dLLM finetuned from LLaDA-8B-Instruct using the token-efficient training algorithm of Arriola & Kuleshov (2026) (detailed recipe in Appendix C.1.2); we refer to this finetuned model as any-order causal LLaDA. We then compare d2-AnyOrder against diffu-GRPO on GSM8K, evaluating test-set accuracy across fixed intervals of a constant total compute budget (measured in FLOPs).

**Results.** As illustrated in Figure 6, d2-AnyOrder consistently improves GSM8K test accuracy within the allocated compute budget, whereas diffu-GRPO fails to achieve mean-

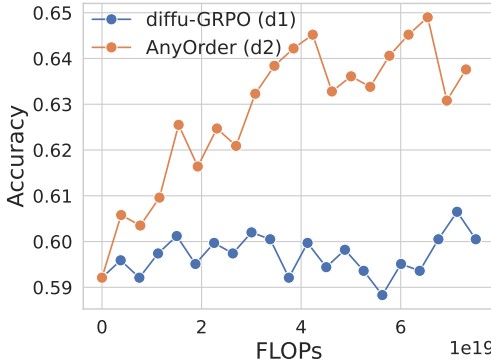

*Figure 6.* Performance-compute dynamics of d2-AnyOrder and diffu-GRPO on any-order causal LLaDA.

ingful gains. This stark contrast highlights the critical importance of exact trajectory likelihood estimation for successful RL in diffusion-based language models.

### 5.2. d2-StepMerge

**Experimental Setup.** To test d2-StepMerge's effect on improving DLM's reasoning capacity, we follow Zhao et al. (2025) in using LLaDA-8B-Instruct (Nie et al., 2025) as the base model and apply different RL post-training algorithms on top of it for four reasoning tasks, including two mathematical reasoning benchmarks (GSM8K (Cobbe et al., 2021) and MATH500 (Lightman et al., 2024)), and two logical reasoning benchmarks (Countdown and Sudoku). For both training and evaluation, two tokens are generated at each time step. We use a group size of 6, and each batch contains 16 questions. Consistent with the protocol in Section 5.1, we maintain a constant compute budget (measured in FLOPs) and evaluate the model's test set performance at fixed FLOP intervals. See more details in Appendix C.2.

**Baselines.** To benchmark our proposed framework, we compare d2 against a diverse set of baselines. Specifically, we include the original LLaDA, LLaDA 1.5 (Zhu et al., 2025), a low-variance direct preference optimization (Rafailov et al., 2023) post-training algorithm, d1 (Zhao et al., 2025), a hybrid method that combines supervised finetuning on the s1k (Muennighoff et al., 2025) long chain-of-thought data with diffu-GRPO, as well as wd1 (Tang et al., 2026), which reformulates policy optimization as a weighted likelihood objective to eliminate the dependence on policy ratios.

**Results.** Shown in Figure 7, d2-StepMerge consistently outperforms diffu-GRPO in all four benchmarks. In Sudoku, Countdown, and GSM8K, d2-StepMerge significantly dominates diffu-GRPO, and in MATH500, d2-StepMerge demonstrates a better trend. These results indicate that d2 achieves a superior trade-off between efficiency and performance. Moreover, Table 6 shows that even without supervised finetuning on extra chain-of-thought data, d2 can outperform existing diffusion reasoning frameworks, demonstrating the

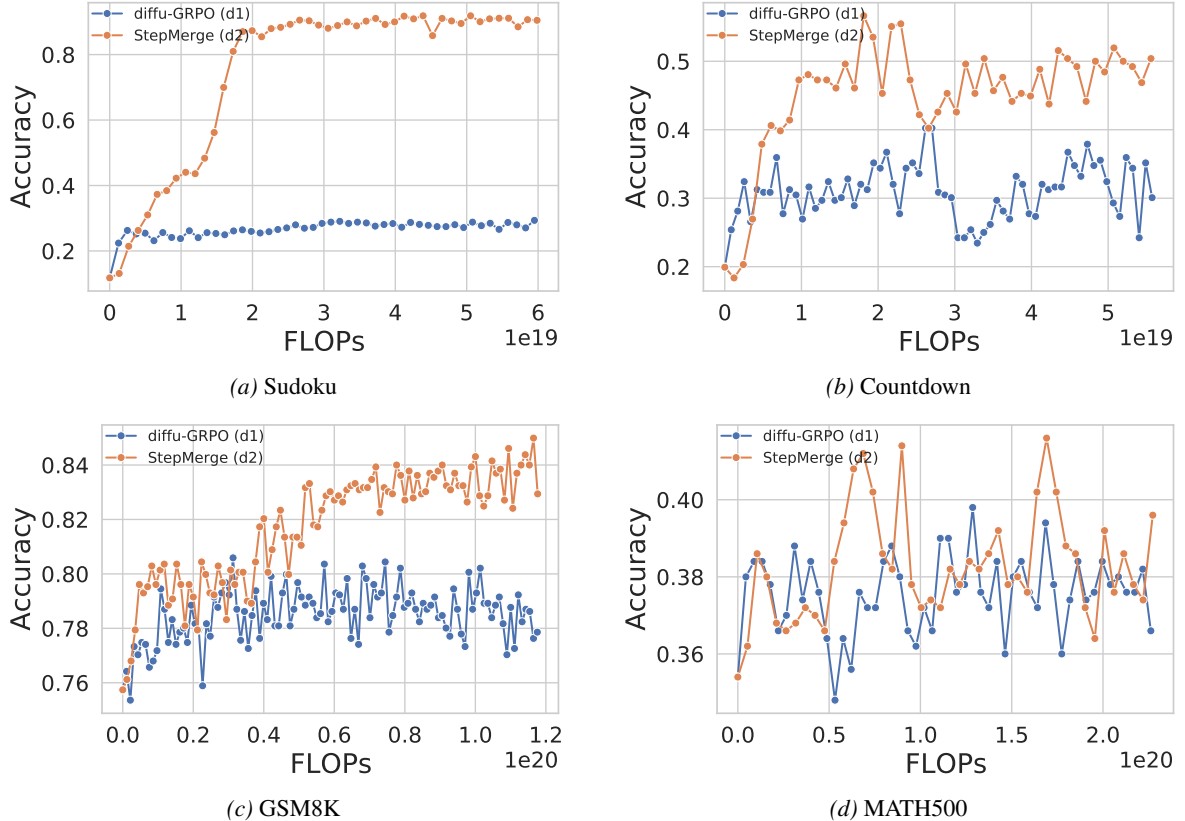

*Figure 7.* Performance-compute dynamics of d2-StepMerge on four reasoning benchmarks.

efficacy of our proposed likelihood estimator.

*Table 6.* Benchmark performance of different reasoning frameworks. † indicates results evaluated on the released checkpoint. ‡ indicates results taken from the corresponding paper. Baselines that include post-training are shaded. Best results are **bolded**.

| Method | Sudoku | Countdown | GSM8K | MATH500 |
|---|---|---|---|---|
| LLaDA[†] | 11.8% | 19.9% | 75.7% | 35.4% |
| LLaDA 1.5[†] | 12.5% | 23.4% | 78.6% | 36.8% |
| d1[‡] | 22.1% | 42.2% | 82.1% | 40.2% |
| wd1[‡] | 25.2% | 51.2% | 82.3% | 39.0% |
| d2 (ours) | **91.9%** | **56.6%** | **85.0%** | **41.6%** |

**Ablation: Varying N in d2-StepMerge.** In Figure 8, we show the Accuracy-FLOP trade-off of d2-StepMerge with different values of $N$ in the Sudoku benchmark. With a small $N$, the model does not reliably converge to competitive performance because the corresponding likelihood estimation is highly inaccurate. In contrast, excessively large $N$ leads to over-allocation of compute to likelihood estimation, resulting in slower convergence. Our results identify $N = 16$ as a favorable balance: it achieves performance comparable to $N = 32$ and $N = 64$, while requiring substantially fewer FLOPs.

**Generalization to Other Architectures.** To further

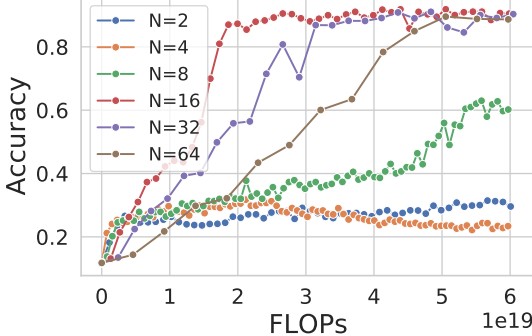

*Figure 8.* d2-StepMerge performance in Sudoku with different $N$.

demonstrate d2-StepMerge's generalization across different masked DLMs, we additionally evaluate on Dream 7B (Ye et al., 2025), a different masked DLM finetuned from Qwen2.5-7B with fully bidirectional attention. All results are evaluated 0-shot for consistency with our other experiments (Dream's original paper uses 8-shot evaluation). As shown in Table 7, d2-StepMerge improves over diffu-GRPO by +12% on Sudoku and +7% on GSM8K, mirroring the pattern observed on LLaDA.

**Coding Benchmarks.** To further demonstrate d2-StepMerge's generalization across different tasks, we evaluate on HumanEval (Chen et al., 2021) and

*Table 7.* Generalization to Dream 7B (0-shot evaluation). Best results are **bolded**.

| Method | Sudoku | GSM8K |
|---|---|---|
| Dream 7B | 0.14 | 0.71 |
| + diffu-GRPO | 0.79 | 0.73 |
| + d2-StepMerge | **0.91** | **0.80** |

MBPP (Austin et al., 2021b). Following Zhao et al. (2025)'s setup—using the same reward function, training set, and hyperparameters—we only replace the training loss with d2-StepMerge ($N = 16$). Both diffu-GRPO and d2-StepMerge are trained at sequence length $L = 256$ and evaluated at $L = 256$ (training length) and $L = 512$ (length generalization). We report the best-checkpoint performance under matched FLOPs.

*Table 8.* Coding accuracy with LLaDA-8B-Instruct as the base model. Best results are **bolded**.

| Method | HumanEval | | MBPP | |
|---|---|---|---|---|
| | $L$=256 | $L$=512 | $L$=256 | $L$=512 |
| LLaDA | 35.3 | 37.8 | 41.2 | 40.4 |
| + diffu-GRPO | **39.0** | 34.8 | 45.5 | 41.6 |
| + d2-StepMerge | **39.0** | **40.9** | **45.9** | **45.5** |

At the training length ($L = 256$), d2-StepMerge matches or modestly improves over diffu-GRPO. The gap widens at $L = 512$: diffu-GRPO regresses on both benchmarks, while d2-StepMerge maintains or improves. This suggests that more accurate likelihood estimation improves the policy's ability to generalize to longer outputs.

## 6. Related Work, Discussion, and Conclusion

**RL for DLMs.** Diffusion language models (Sahoo et al., 2024; 2025; Nie et al., 2025; Ye et al., 2025) have recently emerged as a compelling alternative to AR, sparking research into reinforcement learning tailored for diffusion-based generation. A primary challenge in this domain is to accurately estimate the token likelihood for policy gradient signals. diffu-GRPO (Zhao et al., 2025) approximates this likelihood using logits derived from an all-mask token input. While computationally efficient, this approach introduces significant bias, leading to suboptimal policy updates. To refine this, DiffuCoder's coupled-GRPO (Gong et al., 2026) utilizes the MDLM Evidence Lower Bound (ELBO) (Sahoo et al., 2024) as a proxy for the trajectory likelihood. However, as the ELBO is only a variational bound, it introduces approximation errors that can destabilize RL training. Similarly, wd1 (Tang et al., 2026) reformulates policy optimization as a weighted likelihood objective but inherits the biased estimation method of diffu-GRPO. While LLaDOU (Huang et al., 2025) achieves a more precise decomposition

of likelihood across diffusion timesteps, its full decomposition approach is computationally prohibitive for large-scale applications. Most recently, TraceRL (Wang et al., 2026) proposes merging timesteps for trajectory evaluation; however, unlike our proposed d2-StepMerge, their heuristic approach lacks the rigorous theoretical justification required for consistent likelihood estimation. Concurrent with our work, AGRPO (Zhan, 2026) also uses multiple model forward passes to estimate trajectory likelihood, but does so via random Monte Carlo sampling; d2-StepMerge instead employs a deterministic estimator with an analytically bounded bias (Theorem 4.1).

**Limitations and Future Work.** While d2-AnyOrder provides exact one-pass likelihood estimation, it requires the underlying model to support any-order decoding, which most current masked DLMs do not satisfy by default. Recent any-order dLLMs (Sahoo et al., 2026; Arriola & Kuleshov, 2026) natively support any-order decoding and attain strictly better speed-quality Pareto frontiers than standard masked DLMs. As such models mature, d2-AnyOrder's applicability will expand correspondingly, and we view pursuing a stronger general-purpose any-order dLLM as a natural next step.

**Conclusion.** This paper presents d2, a principled RL framework for diffusion language models. We derive a GRPO-style policy gradient formulation for masked DLMs that depends on accurate trajectory likelihood estimates. Because evaluating these likelihoods is computationally expensive, we develop a family of estimators tailored to distinct model classes. For DLMs that support a sampling algorithm called any-order decoding (AO-dLLMs), we propose d2-AnyOrder, which provides unbiased trajectory likelihood estimates in a single model pass. For standard masked diffusion models, we propose d2-StepMerge, which approximates the trajectory likelihood with multiple forward passes and an analytically bounded error. Empirically, d2 achieves superior performance compared to widely used RL baselines on DLMs that support any-order decoding, and demonstrates state-of-the-art performance on four math and logical reasoning benchmarks, without relying on supervised chain-of-thought finetuning.

## Acknowledgements

We gratefully acknowledge **the Blaise team** (https://blaise.tech/) for their generous provision of high-performance compute resources, which were instrumental in training our diffusion language models. In addition, we thank the Blaise team for valuable technical discussions and hands-on support with both the conceptual development and practical implementation of key components of this work.

This work was partially funded by the National Science Foundation under awards DGE-1922551 and CAREER awards 2046760 and 2145577, and by the National Institute of Health under award MIRA R35GM151243. This research was also supported in part through the use of computational resources provided by Lambda (lambda.ai) in partnership with Open Athena AI Foundation, Inc. We gratefully acknowledge their generous GPU infrastructure grants that helped make this work possible.

## Impact Statement

This work introduces a principled reinforcement learning framework for diffusion language models, improving the reliability of reasoning-oriented post-training and broadening the practical utility of dLLMs.

We acknowledge dual-use considerations specific to this work. Our toxicity steering experiments deliberately illustrate that the same likelihood-estimation machinery used to improve reasoning RL can also be used to steer models toward producing toxic outputs. As with other advances in reasoning-oriented post-training, our methods could be applied with adversarial reward signals to amplify harmful generation. Practitioners building on this work should pair it with appropriate alignment, safety evaluation, and access controls before deployment.

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

# Contents

## A. Theoretical Results

### A.1. Proof of Theorem 3.1

*Proof.* We begin by recalling the definition of the RL objective:

$$\mathcal{J}(\theta) = \mathbb{E}_{\mathbf{x}_0^{1:L} \sim \pi_\theta(\mathbf{x}_0^{1:L}|\mathbf{q})} \big[ r(\mathbf{x}_0^{1:L}, \mathbf{q}) \big]. \tag{11}$$

Taking the policy gradient and expanding the expectation over the trajectory distribution, we have

$$\begin{aligned}
\nabla_\theta \mathcal{J}(\theta) &= \nabla_\theta \sum_{\mathbf{x}_0^{1:L}} \pi_\theta(\mathbf{x}_0^{1:L} \mid \mathbf{q}) \, r(\mathbf{x}_0^{1:L}, \mathbf{q}) \\
&= \nabla_\theta \sum_{\mathbf{x}_0^{1:L}} \sum_{\mathbf{x}_{1:T}^{1:L}} \pi_\theta(\mathbf{x}_0^{1:L} \mid \mathbf{x}_{1:T}^{1:L}, \mathbf{q}) \, \pi_\theta(\mathbf{x}_{1:T}^{1:L} \mid \mathbf{q}) \, r(\mathbf{x}_0^{1:L}, \mathbf{q}) \\
&= \nabla_\theta \sum_{\mathbf{x}_{0:T}^{1:L}} \pi_\theta(\mathbf{x}_{0:T}^{1:L} \mid \mathbf{q}) \, r(\mathbf{x}_0^{1:L}, \mathbf{q}) \\
&= \nabla_\theta \, \mathbb{E}_{\mathbf{x}_{0:T}^{1:L} \sim \pi_\theta(\mathbf{x}_0^{1:L}|\mathbf{q})} \big[ r(\mathbf{x}_0^{1:L}, \mathbf{q}) \big].
\end{aligned} \tag{12}$$

Introducing importance sampling with respect to the stale policy $\pi_{\text{old}}$, we obtain

$$\nabla_\theta \mathcal{J}(\theta) = \nabla_\theta \, \mathbb{E}_{\mathbf{x}_{0:T}^{1:L} \sim \pi_{\text{old}}(\mathbf{x}_{0:T}^{1:L}|\mathbf{q})} \left[ r(\mathbf{x}_0^{1:L}, \mathbf{q}) \, \frac{\pi_\theta(\mathbf{x}_{0:T}^{1:L} \mid \mathbf{q})}{\pi_{\text{old}}(\mathbf{x}_{0:T}^{1:L} \mid \mathbf{q})} \right]. \tag{13}$$

By exploiting the Markov factorization of the backward process, the joint distribution decomposes as

$$\pi_\theta(\mathbf{x}_{0:T}^{1:L} \mid \mathbf{q}) = \pi(\mathbf{x}_T^{1:L}) \prod_{t=0}^{T-1} \pi_\theta(\mathbf{x}_t^{1:L} \mid \mathbf{x}_{t+1}^{1:L}, \mathbf{q}). \tag{14}$$

Substituting this factorization, we obtain

$$\nabla_\theta \mathcal{J}(\theta) = \mathbb{E}_{\mathbf{x}_{0:T}^{1:L} \sim \pi_{\text{old}}(\mathbf{x}_{0:T}^{1:L}|\mathbf{q})} \left[ r(\mathbf{x}_0^{1:L}, \mathbf{q}) \, \nabla_\theta \, \frac{\prod_{t=0}^{T-1} \pi_\theta(\mathbf{x}_t^{1:L} \mid \mathbf{x}_{t+1}^{1:L}, \mathbf{q})}{\prod_{t=0}^{T-1} \pi_{\text{old}}(\mathbf{x}_t^{1:L} \mid \mathbf{x}_{t+1}^{1:L}, \mathbf{q})} \right]. \tag{15}$$

In MDLM, we assume independence across token positions, the expression further decomposes as

$$\nabla_\theta \mathcal{J}(\theta) = \mathbb{E}_{\mathbf{x}_{0:T}^{1:L} \sim \pi_{\text{old}}(\mathbf{x}_{0:T}^{1:L}|\mathbf{q})} \left[ r(\mathbf{x}_0^{1:L}, \mathbf{q}) \, \nabla_\theta \prod_{t=0}^{T-1} \prod_{l=1}^{L} \frac{\pi_\theta(\mathbf{x}_t^l \mid \mathbf{x}_{t+1}^{1:L}, \mathbf{q})}{\pi_{\text{old}}(\mathbf{x}_t^l \mid \mathbf{x}_{t+1}^{1:L}, \mathbf{q})} \right]. \tag{16}$$

Expanding the gradient of the product yields

$$\nabla_\theta \mathcal{J}(\theta) = \mathbb{E}_{\mathbf{x}_{0:T}^{1:L} \sim \pi_{\text{old}}(\mathbf{x}_{0:T}^{1:L}|\mathbf{q})} \left[ r(\mathbf{x}_0^{1:L}, \mathbf{q}) \sum_{t=0}^{T-1} \sum_{l=1}^{L} \nabla_\theta \frac{\pi_\theta(\mathbf{x}_t^l \mid \mathbf{x}_{t+1}^{1:L}, \mathbf{q})}{\pi_{\text{old}}(\mathbf{x}_t^l \mid \mathbf{x}_{t+1}^{1:L}, \mathbf{q})} \prod_{\substack{t' \neq t \\ l' \neq l}} \frac{\pi_\theta(\mathbf{x}_{t'}^{l'} \mid \mathbf{x}_{t'+1}^{1:L}, \mathbf{q})}{\pi_{\text{old}}(\mathbf{x}_{t'}^{l'} \mid \mathbf{x}_{t'+1}^{1:L}, \mathbf{q})} \right]. \tag{17}$$

When evaluated at $\theta = \theta_{\text{old}}$, the product term reduces to 1, yielding

$$\nabla_\theta \mathcal{J}(\theta) \Big|_{\theta=\theta_{\text{old}}} = \mathbb{E}_{\mathbf{x}_{0:T}^{1:L} \sim \pi_{\text{old}}(\mathbf{x}_{0:T}^{1:L}|\mathbf{q})} \left[ r(\mathbf{x}_0^{1:L}, \mathbf{q}) \nabla_\theta \sum_{t=0}^{T-1} \sum_{l=1}^{L} \frac{\pi_\theta(\mathbf{x}_t^l \mid \mathbf{x}_{t+1}^{1:L}, \mathbf{q})}{\pi_{\text{old}}(\mathbf{x}_t^l \mid \mathbf{x}_{t+1}^{1:L}, \mathbf{q})} \right]. \tag{18}$$

Finally, since the model $\pi_\theta$ is only invoked when a masked token $\mathbf{m}$ becomes unmasked, the objective simplifies to

$$\nabla_\theta \mathcal{J}(\theta) \Big|_{\theta=\theta_{\text{old}}} = \nabla_\theta \, \mathbb{E}_{\mathbf{x}_{0:T}^{1:L} \sim \pi_{\text{old}}(\mathbf{x}_{0:T}^{1:L}|\mathbf{q})} \left[ r(\mathbf{x}_0^{1:L}, \mathbf{q}) \sum_{t=0}^{T-1} \sum_{l=1}^{L} \mathbf{1}_{\{\mathbf{x}_{t+1}^l = \mathbf{m}, \, \mathbf{x}_t^l \neq \mathbf{m}\}} \frac{\pi_\theta(\mathbf{x}_t^l \mid \mathbf{x}_{t+1}^{1:L}, \mathbf{q})}{\pi_{\text{old}}(\mathbf{x}_t^l \mid \mathbf{x}_{t+1}^{1:L}, \mathbf{q})} \right]. \tag{19}$$

$\square$

## A.2. Proof of Theorem 4.1

Our proof proceeds in three key steps:

1. **Decompose the diffusion process**: We factor each diffusion step into timing (which tokens unmask) and value (what values they take) components, exploiting the conditional independence structure.

2. **Bound consecutive timesteps**: For adjacent timesteps, we prove the timing component contributes at most $2k \cdot \log 2$ bits (where $k$ tokens unmask), while the value component vanishes under mild assumptions.

3. **Extend to full trajectory**: We aggregate bounds over $N$ blocks, showing each block contributes at most $k_n \log B$ bits, yielding the final $O(U \log(T/N))$ bound.

To keep notation concise, we define a sequence *without* a superscript as $x_t = x_t^{1:L} = [x_t^1, \dots x_t^L]$ and drop the prompt $q$ that we condition on $\pi_\theta(\cdot \mid \cdot, q)$.

### A.2.1. TIMING AND VALUE FACTORIZATION

The reverse process, $\pi_\theta(x_t \mid x_{t+1})$, can be decomposed into two conceptual and computational steps: a *timing* decision of whether to unmask a token, followed by a *value* assignment of what it becomes. This allows us to factor the distribution into two simpler components.

**Definition A.1** (Timing and Value Decomposition). We introduce an indicator variable $S_t^l = \mathbf{1}[x_{t+1}^l = \mathbf{m} \wedge x_t^l \neq \mathbf{m}]$ for the unmasking event, and a categorical random variable $V_t^l = x_t^l$ for the token's value. They are aggregated as $S_t = [S_t^1 \dots S_t^L]$ and $V_t = [V_t^1 \dots V_t^L]$ and factorize the reverse process:

$$\pi_\theta(x_t \mid x_{t+1}) = \underbrace{\tau(S_t \mid x_{t+1})}_{\text{Timing}} \cdot \underbrace{\nu_\theta(V_t \mid S_t^l, x_{t+1})}_{\text{Value}}. \tag{20}$$

The reverse processes decomposes over tokens as $\pi_\theta(x_t \mid x_{t+1}) = \prod_{l=1}^{L} \pi_\theta(x_t^l \mid x_{t+1}^{1:L})$ which emits a similar per-token factorization $\pi_\theta(x_t^l \mid x_{t+1}^{1:L}) = \tau(S_t^l \mid x_{t+1}^l) \cdot \nu_\theta(V_t^l \mid S_t^l, x_{t+1}^{1:L})$. We now define $\tau$ and $\nu_\theta$ as follows.

**Timing Distribution ($\tau$):** This distribution models the probability of the unmasking event $S_t^l$ for any unmasking schedule (e.g. greedy, ancestral, top-k, etc.).

- If a token at $t+1$ is already unmasked ($x_{t+1}^l \neq \mathbf{m}$), it cannot unmask again; thus, the event $S_t^l = 1$ has zero probability.

- If a token is masked ($x_{t+1}^l = \mathbf{m}$), it unmasks with probability $\alpha_t$ determined by the unmasking schedule.

$$\tau(S_t^l = s \mid x_{t+1}^l) = \begin{cases} \alpha_t & \text{if } s = 1 \text{ and } x_{t+1}^l = \mathbf{m} \\ 1 - \alpha_t & \text{if } s = 0 \text{ and } x_{t+1}^l = \mathbf{m} \\ 0 & \text{if } s = 1 \text{ and } x_{t+1}^l \neq \mathbf{m} \\ 1 & \text{if } s = 0 \text{ and } x_{t+1}^l \neq \mathbf{m} \end{cases} \tag{21}$$

**Value Distribution ($\nu_\theta$):** This distribution assigns a value $V_t^l$ to the token, conditional on the timing decision $S_t^l$ and the full sequence context $x_{t+1}$.

- If the decision was to unmask ($S_t^l = 1$), $\nu_\theta$ is the categorical distribution over the vocabulary $\mathcal{V}$ given by the softmax output of the neural network $f_\theta$.

- If the decision was to not unmask ($S_t^l = 0$), the process is deterministic: the token's state from $t + 1$ is simply preserved at time $t$.

$$\nu_\theta(V_t^l = v \mid S_t^l, x_{t+1}) = \begin{cases} \text{softmax}(f_\theta(x_{t+1})_l)_v & \text{if } S_t^l = 1 \\ \delta_{v, x_{t+1}^l} & \text{if } S_t^l = 0 \end{cases} \tag{22}$$

where $\delta_{a,b}$ is the Kronecker delta, enforcing the deterministic state preservation when $S_t^l = 0$.

### A.2.2. CONSECUTIVE TIMESTEP BOUNDS

We begin by analyzing the error over two consecutive timesteps $t$ and $t + 1$. This allows for the simplest possible analysis between the full trajectory and stepmerge trajectory:

$$p_{\text{true}}(x_t, x_{t+1} \mid x_{t+2}) = \pi_\theta(x_t \mid x_{t+1})\pi_\theta(x_{t+1} \mid x_{t+2}) \quad \text{(Full Trajectory)} \tag{23}$$

$$p_{\text{approx}}(x_t, x_{t+1} \mid x_{t+2}) = \pi_\theta(x_t \mid x_{t+2})\pi_\theta(x_{t+1} \mid x_{t+2}) \quad \text{(StepMerge Trajectory)}. \tag{24}$$

We seek to bound the KL divergence between them:

$$D_{\text{KL}}(p_{\text{true}} \| p_{\text{approx}}) = \mathbb{E}_{p_{\text{true}}}\left[\log \frac{p_{\text{true}}(x_t, x_{t+1} \mid x_{t+2})}{p_{\text{approx}}(x_t, x_{t+1} \mid x_{t+2})}\right] = \mathbb{E}_{p_{\text{true}}}\left[\log \frac{\pi_\theta(x_t \mid x_{t+1})}{\pi_\theta(x_t \mid x_{t+2})}\right] \tag{25}$$

**Lemma A.2** (Timing and Value Decomposition of KL Divergence). *The KL divergence between the true and approximate distributions for consecutive timesteps decomposes into a sum of timing and value components:*

$$D_{KL}(p_{true} \| p_{approx}) = D_{KL, \text{ timing}} + D_{KL, \text{ value}} \tag{26}$$

*where the components are defined as:*

$$D_{KL, \text{ timing}} = \mathbb{E}_{(S_t, x_{t+1}) \sim p_{true}}\left[\sum_{l=1}^{L} \log \frac{\tau(S_t^l \mid x_{t+1})}{\tau(S_t^l \mid x_{t+2})}\right] \tag{27}$$

$$D_{KL, \text{ value}} = \mathbb{E}_{(S_t, V_t, x_{t+1}) \sim p_{true}}\left[\sum_{l=1}^{L} \log \frac{\nu_\theta(V_t^l \mid S_t^l, x_{t+1})}{\nu_\theta(V_t^l \mid S_t^l, x_{t+2})}\right] \tag{28}$$

*Proof.* The proof begins with the simplified expression for the KL divergence from Equation 25 and applies the timing-value factorization of Equation 20. The expectation is over the joint distribution $p_{\text{true}}(x_t, x_{t+1} \mid x_{t+2})$, where $x_t$ comprises the

timing and value variables $(S_t, V_t)$.

$$D_{\text{KL}}(p_{\text{true}} \| p_{\text{approx}}) = \mathbb{E}_{(x_t, x_{t+1}) \sim p_{\text{true}}} \left[ \log \frac{\pi_\theta(x_t \mid x_{t+1})}{\pi_\theta(x_t \mid x_{t+2})} \right] \tag{29}$$

$$= \mathbb{E}_{(S_t, V_t, x_{t+1}) \sim p_{\text{true}}} \left[ \log \frac{\prod_l \tau(S_t^l \mid x_{t+1}) \nu_\theta(V_t^l \mid S_t^l, x_{t+1})}{\prod_l \tau(S_t^l \mid x_{t+2}) \nu_\theta(V_t^l \mid S_t^l, x_{t+2})} \right] \tag{30}$$

$$= \mathbb{E}_{(S_t, x_{t+1}) \sim p_{\text{true}}} \left[ \sum_l \log \frac{\tau(S_t^l \mid x_{t+1})}{\tau(S_t^l \mid x_{t+2})} \right]$$

$$+ \mathbb{E}_{(S_t, V_t, x_{t+1}) \sim p_{\text{true}}} \left[ \sum_l \log \frac{\nu_\theta(V_t^l \mid S_t^l, x_{t+1})}{\nu_\theta(V_t^l \mid S_t^l, x_{t+2})} \right] \tag{31}$$

Note that the timing term does not depend on the value variable $V_t$ (and only on $\tau$). Thus it can be marginalized out from the expectation over $p_{\text{true}}$. $\qquad\square$

We now seek to bound the error in timing $D_{\text{KL, timing}}$ and value separately $D_{\text{KL, value}}$.

**Definition A.3** (Conditional Mutual Information). The conditional mutual information $I(A; B \mid C)$ measures the reduction in uncertainty about random variable $A$ from knowing $B$, when $C$ is also known. It can be defined equivalently in terms of conditional entropy or as a Kullback-Leibler (KL) divergence:

$$I(A; B \mid C) = H(A \mid C) - H(A \mid B, C) = D_{\text{KL}}\big(p(a, b \mid c) \| p(a \mid c) p(b \mid c)\big). \tag{32}$$

**Assumption A.4** (Fixed Unmasking Schedule). We assume a schedule where a fixed number of $k$ tokens are unmasked at each timestep $t$.

**Lemma A.5** (Timing KL Bound). *Under the fixed unmasking schedule, the timing component of the KL divergence is bounded by the entropy of the timing decisions.*

$$D_{\text{KL, timing}} \leq 2k \cdot \log 2 \tag{33}$$

*Proof.* The proof first establishes the equivalence between the timing KL divergence (Equation 27) and conditional mutual information, and then bounds this term using entropy.

First, we show the equivalence by applying the Markov property of the true process (Equation 23) in the numerator:

$$D_{\text{KL, timing}} = \mathbb{E}_{\tau_{\text{true}}} \left[ \log \frac{\tau(S_t \mid x_{t+1})}{\tau(S_t \mid x_{t+2})} \right] = \mathbb{E}_{\tau_{\text{true}}} \left[ \log \frac{\tau(S_t \mid x_{t+1}, x_{t+2})}{\tau(S_t \mid x_{t+2})} \right]. \tag{34}$$

We multiply and divide by $\tau(x_{t+1} \mid x_{t+2})$ to get the conditional mutual information (Theorem A.3)

$$D_{\text{KL, timing}} = \mathbb{E}_{\tau_{\text{true}}} \left[ \log \frac{\tau(S_t | x_{t+1}, x_{t+2})}{\tau(S_t | x_{t+2})} \cdot \frac{\tau(x_{t+1} | x_{t+2})}{\tau(x_{t+1} | x_{t+2})} \right] \tag{35}$$

$$= \mathbb{E}_{\tau_{\text{true}}} \left[ \log \frac{\tau(S_t, x_{t+1} | x_{t+2})}{\tau(S_t | x_{t+2}) \tau(x_{t+1} | x_{t+2})} \right] \tag{36}$$

$$= I(S_t; x_{t+1} \mid x_{t+2}). \tag{37}$$

We bound the conditional mutual information term by relating it to the entropy of the timing decisions, which can be decomposed on a per-token basis.

$$
\begin{aligned}
I(S_t; x_{t+1} \mid x_{t+2}) &= H(S_t \mid x_{t+2}) - H(S_t \mid x_{t+1}, x_{t+2}) && \text{(by definition of mutual information)} \\
&\leq H(S_t \mid x_{t+2}) && \text{(since entropy is non-negative)} \\
&= \sum_{l=1}^{L} H(S_t^l \mid x_{t+2}) && \text{(by cond. independence of tokens)}
\end{aligned}
\tag{38}
$$

To evaluate this sum, we partition the tokens into those that unmask in the $[t, t+2)$ interval and those that do not.

$$\sum_{l=1}^{L} H(S_t^l \mid x_{t+2}) = \sum_{l \in \text{Unmasked}} H(S_t^l \mid x_{t+2}) + \sum_{l \notin \text{Unmasked}} H(S_t^l \mid x_{t+2}) \qquad \text{(by partitioning the sum)}$$

$$\leq \sum_{l \in \text{Unmasked}} \log 2 + 0 \qquad \text{(by bounding each term)}$$

$$= 2k \cdot \log 2 \qquad \text{(by summing over the set)} \qquad (39)$$

The second sum vanishes as its tokens have a deterministic timing decision ($S_t^l = 0$), resulting in zero entropy. The first sum is over the $2k$ tokens that unmask in the interval. For each token, $S_t^l$ is a binary random variable representing the choice of unmasking at time $t$ or $t+1$. The entropy of such a variable is maximized at $\log 2$ under maximum uncertainty (a uniform distribution over the two outcomes). Using this upper bound for each of the $2k$ tokens yields the final result. $\qquad \square$

**Definition A.6** (Value Prediction Sensitivity). Let $\epsilon$ be the maximum per-token, pointwise log-ratio of value probabilities between consecutive timesteps, maximized over all possible values, tokens, and states where an unmasking occurs.

$$\epsilon = \max_{v,l,x_{t+1},x_{t+2}} \log \frac{\nu_\theta(V_t^l = v \mid S_t^l = 1, x_{t+1})}{\nu_\theta(V_t^l = v \mid S_t^l = 1, x_{t+2})} \qquad (40)$$

$$= \max_{v,l,x_{t+1},x_{t+2}} \log \frac{\text{softmax}(f_\theta(x_{t+1})_l)_v}{\text{softmax}(f_\theta(x_{t+2})_l)_v} \qquad (41)$$

$$= \max_{v,l,x_{t+1},x_{t+2}} \left( \underbrace{f_\theta(x_{t+1})_{l,v} - f_\theta(x_{t+2})_{l,v}}_{\text{Logit Difference}} - Z_l(x_{t+1}, x_{t+2}) \right) \qquad (42)$$

for the difference between log-softmax normalizers $Z_l(x_{t+1}, x_{t+2}) = \log \left( \frac{\sum_j \exp\left( f_\theta(x_{t+1})_{l,j} \right)}{\sum_j \exp\left( f_\theta(x_{t+2})_{l,j} \right)} \right)$.

*Remark* A.7 (Interpetation of Value Prediction Sensitivity). The value prediction sensitivity $\epsilon$ provides a worst-case guarantee on the stability of the model's value distribution $\nu_\theta$ between consecutive timesteps. It bounds how much $\nu_\theta$ can change for any token value $v$ at a position $l$ by constraining fluctuations in the underlying neural network's raw logits—namely the difference between $f_\theta(\mathbf{x}_{t+1})_{l,v}$ and $f_\theta(\mathbf{x}_{t+2})_{l,v}$ after normalization. A small $\epsilon$ therefore signifies that the entire logit vector for position $l$ remains relatively constant when the context shifts by one step (from $x_{t+1}$ to $x_{t+2}$). This logit-level stability ensures that the model's predictive distribution is not volatile, thereby validating our StepMerge approximation, which relies on the assumption that intermediate states can be skipped without drastically altering the final trajectory likelihood.

**Lemma A.8** (Value KL Bound). *Under the fixed unmasking schedule, the value component of the KL divergence is bounded in terms of the value prediction sensitivity $\epsilon$.*

$$D_{\text{KL, value}} \leq k \cdot \epsilon \qquad (43)$$

*Proof.* We start with the definition of the value KL divergence:

$$D_{\text{KL, value}} = \mathbb{E}_{p_{\text{true}}} \left[ \sum_{l=1}^{L} \log \frac{\nu_\theta(V_t^l \mid S_t^l, x_{t+1})}{\nu_\theta(V_t^l \mid S_t^l, x_{t+2})} \right] \qquad (44)$$

To analyze the sum inside the expectation, we partition the token indices based on the timing decision $S_t^l$, which indicates whether token $l$ is unmasked at step $t$. Let $\mathcal{U}_t = \{l \mid S_t^l = 1\}$ be the set of indices for tokens that unmask, and $\mathcal{U}_t^c = \{l \mid l \notin \mathcal{U}_t\} = \{l \mid S_t^l = 0\}$ be the complementary set for all other tokens. The sum can be explicitly decomposed as:

$$\sum_{l=1}^{L} \log \frac{\nu_\theta(\dots)}{\nu_\theta(\dots)} = \underbrace{\sum_{l \in \mathcal{U}_t} \log \frac{\nu_\theta(V_t^l \mid S_t^l = 1, x_{t+1})}{\nu_\theta(V_t^l \mid S_t^l = 1, x_{t+2})}}_{\text{Tokens unmasked at time } t} + \underbrace{\sum_{l \in \mathcal{U}_t^c} \log \frac{\nu_\theta(V_t^l \mid S_t^l = 0, x_{t+1})}{\nu_\theta(V_t^l \mid S_t^l = 0, x_{t+2})}}_{\text{Tokens } not \text{ unmasked at time } t} \qquad (45)$$

$$\leq \sum_{l \in \mathcal{U}_t} \epsilon + \sum_{l \in \mathcal{U}_t^c} \log \frac{1}{1} \qquad (46)$$

$$= k \cdot \epsilon \qquad (47)$$

For tokens that are unmasked at time $t$ ($l \in \mathcal{U}_t$), the value distribution $\nu_\theta$ is defined by the model's softmax output, and the log-ratio is bounded by the value prediction sensitivity ($\epsilon$). For any token that is not unmasked at time $t$ ($l \in \mathcal{U}_t^c$), the value-setting process is a deterministic identity transformation, where $V_t^l = x_{t+1}^l$ with probability 1.

Since this upper bound holds for any trajectory, the expectation over all trajectories is also bounded by the same constant. $\square$

**Lemma A.9** (Consecutive Timestep Bound). *Under the fixed unmasking schedule Theorem A.4, the total KL divergence between the true and approximate distributions for consecutive timesteps is bounded by the sum of the timing and value bounds.*

$$D_{KL}(p_{true}\|p_{approx}) \leq 2k \cdot \log 2 + k \cdot \epsilon \tag{48}$$

*Proof.* The result follows directly by combining the bounds from the preceding lemmas.

$$D_{\text{KL}}(p_{\text{true}}\|p_{\text{approx}}) = D_{\text{KL, timing}} + D_{\text{KL, value}} \leq (2k \cdot \log 2) + (k \cdot \epsilon) \tag{49}$$

$\square$

### A.2.3. ENTIRE TRAJECTORY BOUND

We extend the analysis from an $L$-length sequence at consecutive timesteps $x_t, x_{t+1}$ to the entire trajectory $x_0 \ldots x_T$ with $N$ total blocks. To do so, first consider block $n$ with $B = L/N$ timesteps spanning $t = [nB, nB+1, \ldots, nB+(B-1)]$.

Within this block, the true reverse process is a Markov chain over the full trajectory:

$$p_{\text{true}}^{(n)}(x_{nB}, \ldots, x_{nB+(B-1)} \mid x_{nB+B}) = \prod_{t=nB}^{nB+(B-1)} \pi_\theta(x_t \mid x_{t+1}) \quad \text{(Full Trajectory)} \tag{50}$$

The StepMerge approximation, however, assumes each state $x_t$ in the block is generated independently conditioned only on the final state $x_{nB+B}$:

$$p_{\text{approx}}^{(n)}(x_{nB}, \ldots, x_{nB+(B-1)} \mid x_{nB+B}) = \prod_{t=nB}^{nB+(B-1)} \pi_\theta(x_t \mid x_{nB+B}) \quad \text{(StepMerge Trajectory)} \tag{51}$$

**Lemma A.10** (Block KL Bound). *Under the fixed unmasking schedule Theorem A.4, the KL divergence for block $n$ is bounded by:*

$$D_{KL}^{(n)}(p_{true}^{(n)}\|p_{approx}^{(n)}) \leq \frac{kB(B+1)}{2} \log 2 + Bk \cdot \epsilon_{block} \tag{52}$$

*where $\epsilon_{block}$ is the maximum value prediction sensitivity over the timesteps in the block.*

*Proof.* The KL divergence for block $n$ is the expectation of the log-ratio of the true and approximate distributions.

$$D_{\text{KL}}^{(n)} = \mathbb{E}_{p_{\text{true}}^{(n)}} \left[ \log \frac{\prod_{t=nB}^{nB+(B-1)} \pi_\theta(x_t \mid x_{t+1})}{\prod_{t=nB}^{nB+(B-1)} \pi_\theta(x_t \mid x_{nB+B})} \right] \tag{53}$$

$$= \mathbb{E}_{p_{\text{true}}^{(n)}} \left[ \sum_{t=nB}^{nB+(B-1)} \log \frac{\pi_\theta(x_t \mid x_{t+1})}{\pi_\theta(x_t \mid x_{nB+B})} \right] \tag{54}$$

$$= \underbrace{\mathbb{E}_{p_{\text{true}}^{(n)}} \left[ \sum_{t=nB}^{nB+(B-1)} \log \frac{\tau(S_t \mid x_{t+1})}{\tau(S_t \mid x_{nB+B})} \right]}_{D_{\text{KL, timing}}^{(n)}} + \underbrace{\mathbb{E}_{p_{\text{true}}^{(n)}} \left[ \sum_{t=nB}^{nB+(B-1)} \log \frac{\nu_\theta(v_t \mid S_t, x_{t+1})}{\nu_\theta(v_t \mid S_t, x_{nB+B})} \right]}_{D_{\text{KL, value}}^{(n)}} \tag{55}$$

Using the timing-value factorization Equation 20 and linearity of expectation, we decompose into block-level timing and value components, $D_{\text{KL}}^{(n)} = D_{\text{KL, timing}}^{(n)} + D_{\text{KL, value}}^{(n)}$, which we bound separately.

**Timing Bound.** The timing component is the sum of conditional mutual information terms

$$D_{\text{KL, timing}}^{(n)} = \sum_{t=nB}^{nB+(B-1)} \mathbb{E}_{p_{\text{true}}^{(n)}} \left[ \log \frac{\tau(S_t \mid x_{t+1})}{\tau(S_t \mid x_{nB+B})} \right] = \sum_{t=nB}^{nB+(B-1)} I(S_t; x_{t+1} \mid x_{nB+B}) \tag{56}$$

following the derivation in Theorem A.5. We can further bound the conditional mutual information by the conditional entropy from Theorem A.3:

$$I(S_t; x_{t+1} \mid x_{nB+B}) = H(S_t \mid x_{nB+B}) - H(S_t \mid x_{t+1}, x_{nB+B}) \leq H(S_t \mid x_{nB+B}). \tag{57}$$

Due to mutual exclusivity, each token can unmask at most once. For a token $l$ masked at $x_{nB+B}$, define $T_l$ as its unmasking time (if any). The collection $\{S_t^l\}_{t=nB}^{nB+B-1}$ encodes which value $T_l$ takes, where $T_l \in \{nB, ..., nB + B - 1, \infty\}$. Thus:

$$\sum_{t=nB}^{nB+B-1} H(S_t^l \mid x_{nB+B}) \leq H(T_l \mid x_{nB+B}^l = \mathbf{m}) \leq \log(B+1) \tag{58}$$

Since exactly $kB$ tokens unmask during the block:

$$D_{\text{KL, timing}}^{(n)} \leq \sum_{l \in \mathcal{M}_{nB+B}} H(T_l \mid x_{nB+B}^l = \mathbf{m}) \leq kB \cdot \log(B+1) \tag{59}$$

where $\mathcal{M}_{nB+B} = \{l : x_{nB+B}^l = \mathbf{m}\}$ is the set of masked tokens at the block end.

**Value Bound.** The value component of the divergence is:

$$D_{\text{KL, value}}^{(n)} = \sum_{t=nB}^{nB+(B-1)} \mathbb{E}_{p_{\text{true}}^{(n)}} \left[ \sum_{l=1}^{L} \log \frac{\nu_\theta(V_t^l \mid S_t^l, x_{t+1})}{\nu_\theta(V_t^l \mid S_t^l, x_{nB+B})} \right] \tag{60}$$

Following Theorem A.6, we define a block-level value prediction sensitivity $\epsilon_{block}$ as the maximum log-ratio within the block:

$$\epsilon_{block} = \max_{t \in [nB,(n+1)B), v, l, x_{t+1}} \log \frac{\nu_\theta(V_t^l = v \mid S_t^l = 1, x_{t+1})}{\nu_\theta(V_t^l = v \mid S_t^l = 1, x_{nB+B})} \tag{61}$$

At each timestep $t$, the inner sum over tokens $l$ is non-zero only for the $k$ tokens being unmasked ($S_t^l = 1$) by definition of $\nu_\theta$ (Equation 22). For each of these, the log-ratio is bounded by definition by $\epsilon_{block}$. Therefore, the entire sum inside the expectation is bounded by $k \cdot \epsilon_{block}$. Summing over the $B$ timesteps yields the total value bound:

$$D_{\text{KL, value}}^{(n)} \leq \sum_{t=nB}^{nB+(B-1)} k \cdot \epsilon_{block} = Bk \cdot \epsilon_{block} \tag{62}$$

**Total Bound.** Combining the bounds for the timing and value components gives the final result for the block-level KL divergence.

$$D_{\text{KL}}^{(n)} = D_{\text{KL, timing}}^{(n)} + D_{\text{KL, value}}^{(n)} \leq kB \cdot \log(B+1) + Bk \cdot \epsilon_{block} \tag{63}$$

$\square$

Finally, we aggregate the per-block errors to establish a bound for the entire generation trajectory. Since the approximation for each block is conditionally independent of the others given the state at the end of the block, the total KL divergence is the sum of the per-block KL divergences.

**Theorem A.11** (Main Bound). *Let $L$ be the total number of tokens (each unmasked exactly once), $B = T/N$ be the number of timesteps per block, and $\epsilon_{block}$ be the maximum value prediction sensitivity within a block. The KL divergence between the true sequential process (Full Trajectory) and the block-parallel approximation (StepMerge Trajectory) is bounded by:*

$$D_{KL}(p_{true} \| p_{approx}) \leq L \cdot \log(B+1) + L \cdot (B-1) \cdot \epsilon_{block} \tag{64}$$

*Proof.* For block $n$, let $k$ be the number of tokens unmasked per timestep in that block. From the tighter timing bound using mutual exclusivity and the value bound:

$$D_{\text{KL}}^{(n)} \leq kB \cdot \log(B+1) + kB \cdot \epsilon_{block} \tag{65}$$

Summing over all $N$ blocks:

$$D_{\text{KL}} = \sum_{n=0}^{N-1} D_{\text{KL}}^{(n)} \leq \sum_{n=0}^{N-1} \left( kB \cdot \log(B+1) + kB \cdot \epsilon_{block} \right) \tag{66}$$

$$= \left( \sum_{n=0}^{N-1} kB \right) \log(B+1) + \left( \sum_{n=0}^{N-1} kB \right) \epsilon_{block} \tag{67}$$

Since each token unmasks exactly once and there are $L$ tokens total we know $\sum_{n=0}^{N-1} kB = L$. This yields:

$$D_{\text{KL}} \leq L \cdot \log(B+1) + L \cdot \epsilon_{\text{block}} \tag{68}$$

Substituting $B = T/N$:

$$D_{\text{KL}} \leq L \cdot \log\left( \frac{T}{N} + 1 \right) + L \cdot \epsilon_{\text{block}} \tag{69}$$

$\square$

## B. d2 Details

### B.1. KL Divergence Estimation

Following Zhao et al. (2025), we utilize the following estimator to estimate the KL divergence in the d2 loss function.

$$D_{\text{KL}} \left( \pi_\theta(\mathbf{x}^{1:L} \mid \mathbf{q}) \,\|\, \pi_{\text{ref}}(\mathbf{x}^{1:L} \mid \mathbf{q}) \right) \approx \sum_{l=1}^{L} \frac{\pi_{\text{ref}}(\mathbf{x}^l \mid \mathbf{q})}{\pi_\theta(\mathbf{x}^l \mid \mathbf{q})} - 1 - \log \frac{\pi_{\text{ref}}(\mathbf{x}^l \mid \mathbf{q})}{\pi_\theta(\mathbf{x}^l \mid \mathbf{q})}, \tag{70}$$

where $\pi(\mathbf{x}^l \mid \mathbf{q})$ comes from the corresponding trajectory likelihood estimator.

### B.2. Any-Order Decoding with Prompts

In Algorithm 2, we describe the prompted version of any-order decoding, which is practically used on the reasoning benchmark experiments.

---

**Algorithm 2** Any-Order Decoding with Prompts

---

**Input:** DLM model $p_\theta$, sequence length $L$, prompt $\mathbf{q}^{1:L_q}$.
$\mathbf{x}^{L_q+1:L_q+L} \leftarrow \mathbf{m}^{L_q+1:L_q+L}; \sigma(L_q + 1), \ldots, \sigma(L_q + L) \leftarrow L_q + L + 1; \sigma(1), \ldots, \sigma(L_q) \leftarrow L_q; n \leftarrow L_q$
**while** $n < L$ **do**
    **for** $l = 1$ to $L$ **do**
        $\text{attn}(\mathbf{x}^{\sigma(l)}) \leftarrow \mathbf{x}_0^{\sigma(\leq l)}$ if $\mathbf{x}^{\sigma(l)} \neq \mathbf{m}$ else $\mathbf{x}_0^{\sigma(<l)} \cup \mathbf{m}^{\sigma(l)}$
    **end for**
    $\mathbf{x}_0^{l_{1:k}} \sim p_\theta(\cdot \mid \mathbf{q}^{q:L_q} \oplus \mathbf{x}^{L_q+1:L_q+L}, \text{attn})$
    $\mathbf{x}^{l_{1:k}} \leftarrow \mathbf{x}_0^{l_{1:k}}; \sigma(l_j)_{j=1}^k \leftarrow n + j; n \leftarrow n + k$
**end while**
**return** $\mathbf{x}^{1:L}$

---

### B.3. d2-AnyOrder: One-Shot Likelihood Evaluation with Prompts

This section details the one-shot likelihood evaluation for sequences decoded using the prompted any-order decoding algorithm (described in Algorithm 2).

**One-shot Trajectory Likelihood.** Let a sampled trajectory of $L$ tokens be denoted as $\mathbf{x}_{0:T}^{L_q+1:L_q+L}$. Here, the trajectory likelihood,

$$\pi_\theta(\mathbf{x}_{0:T}^{L_q+1:L_q+L} \mid \mathbf{q}^{1:L_q}) = \prod_{l=L_q+1}^{L_q+L} \pi_\theta(\mathbf{x}^{\sigma(l)} \mid \mathbf{x}^{\sigma(<l)}), \tag{71}$$

can be computed in a single model forward pass. To efficiently implement this, we construct an input sequence of length $L_q + 2L$ by concatenating the prompt, the clean tokens, and a set of mask tokens: $\mathbf{q}^{1:L_q} \oplus \mathbf{x}_0^{1:L} \oplus \mathbf{m}^{L+1:2L}$. We assign positional encoding indices $\mathrm{pos}_l$ to ensure that each clean token and its corresponding mask token share the same index:

$$\mathrm{pos}_l = \begin{cases} l, & l \leq L_q + L \\ l - L, & L_q + L < l \leq L_q + 2L. \end{cases} \tag{72}$$

The attention mask is specifically crafted such that:

- A clean token, either $\mathbf{q}^{\sigma(l)}$ or $\mathbf{x}_0^{\sigma(l)}$, attends to $\mathbf{x}_0^{\sigma(\leq l)}$;

- A mask token $\mathbf{m}^{L+\sigma(l)}$ attends to $\mathbf{x}_0^{\sigma(<l)} \cup \mathbf{m}^{L+\sigma(l)}$.

As illustrated in Figure 9, this configuration ensures that the output logits of the mask token at position $L + l$ exactly reproduce the logits of $\mathbf{x}_0^l$ as if it were decoded during sampling.

### B.4. d2 Training Algorithms

We present the pseudocode of d2-StepMerge and d2-AnyOrder in Algorithm 3 and Algorithm 4.

---

**Algorithm 3** d2-StepMerge

---

**Input:** Reward model $r$, reference model $\pi_{\mathrm{ref}}$, prompt distribution $\mathcal{Q}$, number of completions per prompt $G$, number of inner updates $n$, number of time segments $N$.
Initialize $\pi_\theta \leftarrow \pi_{\mathrm{ref}}$
**repeat**
   $\pi_{\mathrm{old}} \leftarrow \pi_\theta$
   Sample prompt $\mathbf{q} \sim \mathcal{Q}$
   Sample $G$ completion trajecotries $\{\mathbf{x}_{0:T}^{(i)}\}_{i=1}^G \sim \pi_{\mathrm{old}}(\cdot \mid \mathbf{q})$
   Compute advantage $\{A_{(i)}\}_{i=1}^G$ (see Section 2.3)
   **for** $j = 1$ to $N$ **do**
      `stop_gradient`(Compute and collect $\{\pi_{\mathrm{old}}(\mathbf{x}_{j\frac{T}{N}:(j+1)\frac{T}{N}}^{(i)} \mid \mathbf{q})\}_{i=1}^G$)
      `stop_gradient`(Compute and collect $\{\pi_{\mathrm{ref}}(\mathbf{x}_{j\frac{T}{N}:(j+1)\frac{T}{N}}^{(i)} \mid \mathbf{q})\}_{i=1}^G$)
   **end for**
   **for** gradient_iterations 1 to $n$ **do**
      **for** $j$ 1 to $N$ **do**
         Compute d2-StepMerge GRPO objective (Eq. (3.2.2)) with respect to $\{\mathbf{x}_{j\frac{T}{N}:(j+1)\frac{T}{N}}^{(i)}\}_{i=1}^G$
         Backward pass to calculate gradient
      **end for**
      Update $\theta$ with optimizer.
   **end for**
**until** converged
**return** $\pi_\theta$

---

## C. Additional Experimental Details

### C.1. d2-AnyOrder

#### C.1.1. Eso-LM

**Hyperparameters.** In this experiment, we reuse the 'Eso-LM-B-alpha-1' checkpoint released by Sahoo et al. (2026), which is the pure diffusion version of Eso-LM without AR integration. In both training and evaluation, we let the model generate

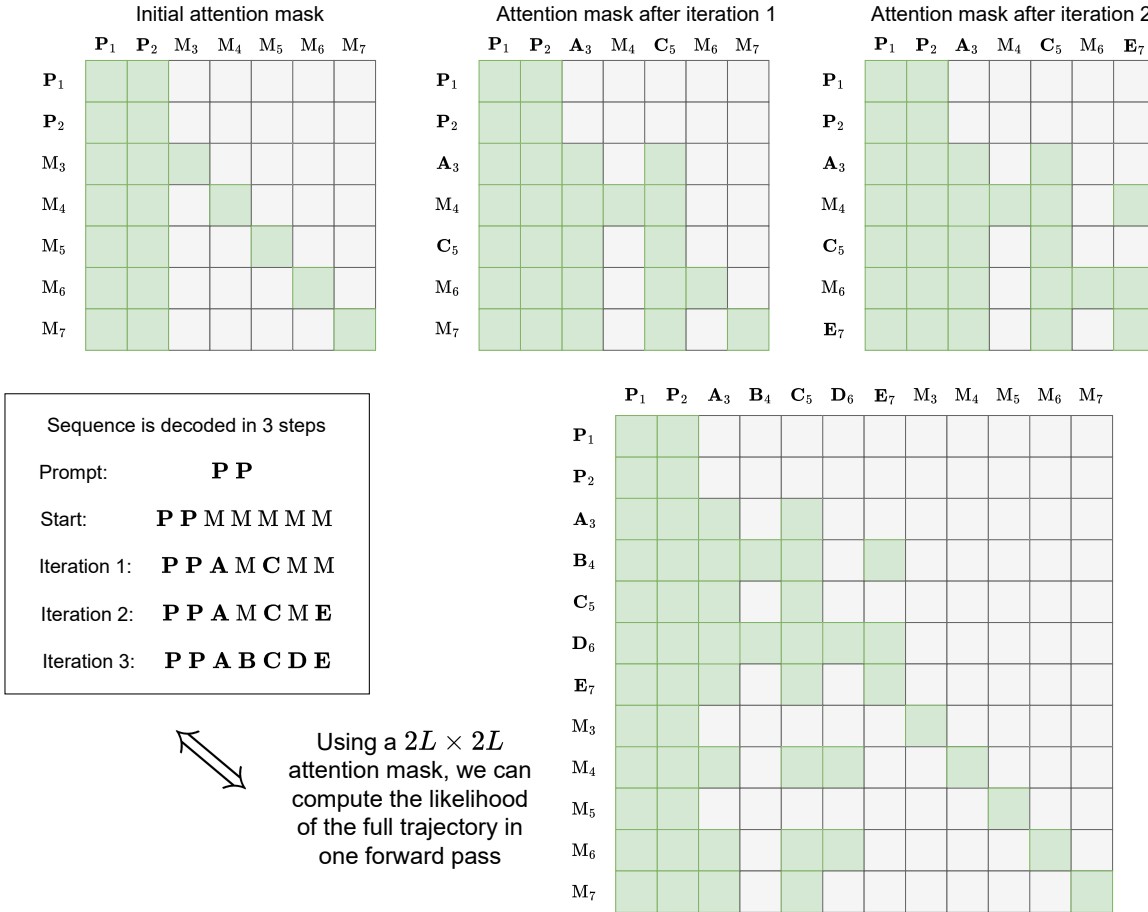

*Figure 9.* Illustration of the attention pattern of an any-order causal AO-ARM when decoding with prompts. Green squares represent attention is 'turned on', i.e., attention bias of 0, between queries (rows) and keys / values (columns), gray represents no attention, i.e., attention bias of $-\infty$. Subscripts denote positional embedding ids. In this example, the sequence is generated as a completion to the prompt **PP** with the following trajectory: $\mathrm{MMMMM} \to \mathbf{A}\mathrm{M}\mathbf{C}\mathrm{MM} \to \mathbf{A}\mathrm{M}\mathbf{C}\mathrm{M}\mathbf{E} \to \mathbf{ABCDE}$. *(Top)* Any-order causal attention mask during generation. *(Bottom)* $(L_q + 2L) \times (L_q + 2L)$ attention mask that enables computation of likelihood for entire sequence trajectory in one forward pass. The model receives a $L_q + 2L$ sequence consisting of a concatenation of the $L$ generated tokens and $L$ mask tokens. The likelihood is computed using the output of the last $L$ output tokens.

---

**Algorithm 4** d2-AnyOrder

---

**Input:** Reward model $r$, reference model $\pi_{\text{ref}}$, prompt distribution $\mathcal{Q}$, number of completions per prompt $G$, number of inner updates $n$.
Initialize $\pi_\theta \leftarrow \pi_{\text{ref}}$
**repeat**
    $\pi_{\text{old}} \leftarrow \pi_\theta$
    Sample prompt $\mathbf{q}^{1:L_q} \sim \mathcal{Q}$
    Sample $G$ completions $\{\mathbf{x}_0^{L_q+1:L_q+L}\}_{i=1}^G \sim \pi_{\text{old}}(\cdot \mid \mathbf{q}^{1:L_q})$
    Compute advantage $\{A_{(i)}\}_{i=1}^G$ (see Section 2.3)
    Build input sequence $INPUT = \mathbf{q}^{1:L_q} \oplus \mathbf{x}_0^{L_q+1:L_q+L} \oplus \mathbf{m}^{L_q+L+1:L_q+2L)}\}_{i=1}^G$
    Build attention mask, see Appendix B.3
    `stop_gradient`(compute $\{\pi_{\text{old}}^{\text{AO}}(\mathbf{x}_0^{L_q+1:L_q+L} \mid INPUT\})$
    `stop_gradient`(compute $\{\pi_{\text{ref}}^{\text{AO}}(\mathbf{x}_0^{L_q+1:L_q+L} \mid INPUT\})$
    **for** gradient_iterations 1 to $n$ **do**
        Compute d2-AnyOrder GRPO objective (Eq. (3.2.2))
        Update $\theta$ with optimizer.
    **end for**
**until** converged
**return** $\pi_\theta$

---

sentences with 512 tokens in free form. We set the group size as 16, and each group consists of a batch of 4 samples. When generating token sequences, we apply no annealing, i.e., the temperature value is set as 1.0.

**Baselines.** Similar to d2, DDPO also decomposes the sequence likelihood along the time steps, and the original version of DDPO (Black et al., 2024) is a PPO-style algorithm. To conduct a fair comparison, we remove the value network in DDPO and use the group advantage instead. In addition, DDPO does not involve trust regions or KL divergence with the reference model, so we also remove the corresponding parts in d2 to conduct a fair comparison.

### C.1.2. ANY-ORDER CAUSAL LLADA: FINETUNE

Our finetuning recipe consists of two stages: data collection and finetuning. In the data collection stage, we aim to collect the sample sequences generated by LLaDA-2.0-mini (Bie et al., 2025) served on the dLLM inference engine dInfer (Ma et al., 2025). In the finetuning stage, we finetune the LLaDA-8B-Instruct model on the LLaDA-2.0-mini distillation data that we collected before using the token-efficient training algorithm from Arriola & Kuleshov (2026).

**Data collection.** First, we collect distillation data from a subset of prompts of the 'neginashz/star-sft-intellect-instruct-3' dataset from Hugging Face (Ashz, 2024). In particular, we collect 150k prompts consisting of 15k prompts from the 'am_chat', 'am_if', 'openreasoning_code', 'openreasoning_science', 'openreasoning_tool', and the 'toucan_tool' splits, and 60k prompts from the 'openreasoning_math' split. We then input these prompts into the LLaDA-2.0-mini model with a maximum generation length of 1024, block size of 32, and confidence threshold of 0.9 (Wu et al., 2026). Second, we collect instruction tuning data from the train set of the corresponding reasoning benchmarks, i.e., GSM8K and MATH500. Similarly, we input the prompts from the train sets of these data into LLaDA-2.0-mini and collect its outputs.

**Any-order causal finetuning algorithm.** We follow Arriola & Kuleshov (2026) to utilize their token-efficient training algorithm to do any-order causal finetuning by useing the AO-ARM objective:

$$\mathcal{L}_{\text{AO-ARM}} = \mathbb{E}_{\sigma \sim U(S_D)} \left[ \sum_{l=1}^L \log p_\theta(\mathbf{x}_0^{\sigma(l)} \mid \mathbf{x}_0^{\sigma(<l)}) \right]. \tag{73}$$

In practice, given a clean token sequence $\mathbf{q}^{1:L_q} \oplus \mathbf{x}^{L_q+1:L_q+L}$ we first randomly sample a permutation $\sigma$ of integer $1, \ldots, L$. Note that here, we slightly tweak $\sigma$ to make it respect block decoding. In other words, the permutation is only applied within each block, and among blocks we use a left-to-right order. After that, we construct a $L_q + 2L$ sequence $\mathbf{q}^{1:L_q} \oplus \mathbf{x}^{L_q+1:L_q+L} \oplus \mathbf{m}^{L_q+L:L_q+2L}$. We then assign positional encoding indices $\text{pos}_l$ to ensure that each clean token and its corresponding mask token share the same index:

$$\text{pos}_l = \begin{cases} l, & l \leq L_q + L \\ l - L, & L_q + L < l \leq L_q + 2L. \end{cases} \tag{74}$$

The attention mask is specifically crafted such that:

- A prompt query $\mathbf{q}^l$ attends to $\mathbf{q}^{1:L_q}$;

- A clean token $\mathbf{x}_0^l$ attends to $\mathbf{q}^{1:L_q} \cup \mathbf{x}_0^{\sigma(\leq l)}$;

- A mask query $\mathbf{m}^{L+l}$ attends to $\mathbf{q}^{1:L_q} \cup \mathbf{x}_0^{\sigma(<l)} \cup \mathbf{m}^{L+\sigma(l)}$.

We then input the toke sequence $\mathbf{q}^{1:L_q} \oplus \mathbf{x}^{L_q+1:L_q+L} \oplus \mathbf{m}^{L_q+L:L_q+2L}$ into the LLaDA model with the crafted attention mask and positional encodings. By computing loss on the logits of the last $L$ mask tokens, we are trarning the LLaDA model into an any-order causal model.

**Any-order causal finetuning setting.** Our finetuning constsits of two stages. The first stage is to finetune the LLaDA-8B-Instruct checkpoint on the Intellect-SFT distillation data that we collected. The second stage is to finetune the checkpoint after the first round of finetuning using the GSM8K / MATH500 distillation data to teach the model how to do instruction following. In both stages, we set aside a separate validation sets from the training set, and pick the checkpoint with the lowest validation loss. In terms of hyperparameters, we use a learning rate of 1e-5, a block size of 32, and a batch size of 8. Unlike Zhao et al. (2025), we do not apply LoRA in the finetuning stage. Instead, we freeze all other parameters and finetune the 'q_proj', 'k_proj', and 'v_proj' layers. In Appendix D.1.2, we provide an ablation results demonstrating the efficacy of the first finetune stage, where the model is exposed to a more diverse and larger set of data.

### C.1.3. ANY-ORDER CAUSAL LLADA: RL

We follow the setting in Zhao et al. (2025) by applying LoRA layers (Hu et al., 2022) to the q_proj, k_proj, v_proj, o_proj, up_proj, down_proj, and gate_proj layers of LLaDA-8B-Instruct with a rank of 128, $\alpha$ of 64, and a dropout rate of 0.05. Unlike Zhao et al. (2025), we do not apply quantization to the model to accelerate training. We also turn off the random masking on prompt tokens for both d2-AnyOrder and diffu-GRPO. We set the learning rate as 1e-6, the number of inner iterations as 2, and group size as 8 to stablize training. $\beta$ is set as 0.04, $\epsilon$ is 0.5, and the effective batch size is 16. The temperature of on-policy sampling is set as 0.9. During evaluation, the sampling temperature is 0, i.e., we apply greedy sampling on the test set. To keep it consistent with the d2-StepMerge experiments, at each time step, two tokens with the largest two confidence scores are decoded simultaneously. To reduce the train-test mismatch, we artificially assign a decoding order to the two tokens simultaneously decoded using a left-to-right order. In other words, when constructing the causal attention mask for the remaining sampling steps, we let the token on the right attend to the token on the left. This sampling setting is consistent across RL training and evaluation. For both training and evaluation, the block size of LLaDA (Nie et al., 2025) is set as 32. In this experiment, we apply diffu-GRPO directly on any-order causal LLaDA checkpoint without SFT. We apply the same reward functions as in Zhao et al. (2025). Unlike Zhao et al. (2025), we keep the standard deviation denominator when computing advantages. For GSM8K, we use a maximal prompt length of 200 and a sequence length of 256. All hyperparameters are shared for both the d2-AnyOrder runs and the diffu-GRPO runs reported in Figure 6 to ensure fair comparison. We also report a new set of experiments on MATH500 in Appendix D.1.2, where we use a generation length of 512.

### C.2. d2-StepMerge

**Hyperparameters.** In this set of experiments, we follow the setting in Zhao et al. (2025) by applying LoRA layers (Hu et al., 2022) to the q_proj, k_proj, v_proj, o_proj, up_proj, down_proj, and gate_proj layers of LLaDA-8B-Instruct with a rank of 128, $\alpha$ of 64, and a dropout rate of 0.05. To reduce GPU memory consumption, we apply 4-bit quantization. We also apply a random mask on the prompt tokens with a probability of 0.15. The learning rate is set as 3e-6, $\beta$ is set as 0.04, and $\epsilon$ is 0.5. For GSM8K, MATH500, and Countdown, the temperature of on-policy sampling is set as 0.9, while the temperature for Sudoku is set as 0.3. During evaluation, the sampling temperature is 0, i.e., we apply greedy sampling on the test set. For both training and evaluation, the block size of LLaDA (Nie et al., 2025) is set as 32. In Figure 7, we apply diffu-GRPO directly on LLaDA-8B-Instruct without SFT, and in Table 6, the d1 results are copied from Zhao et al. (2025), with SFT included. We apply the same reward functions and drop the standard deviation denominator when computing advantages as in Zhao et al. (2025). In Table 9, we report the $N$ values, sequence lengths, inner iteration numbers, and maximum prompt lengths utilized for the four benchmarks. All hyperparameters are shared for both the d2-AnyOrder runs and the diffu-GRPO runs reported in Figure 7 to ensure fair comparison. In the Sudoku ablation, we reuse all the hyperparameter settings reported in 7 and only alter $N$.

*Table 9.* N, sequence length, innter iteration number, and max prompt length applied in the d2-StepMerge experiments.

|  | GSM8K | MATH500 | Countdown | Sudoku |
|---|---|---|---|---|
| Seq. Length | 256 | 512 | 128 | 128 |
| N | 8 | 16 | 16 | 16 |
| Iteration Number | 12 | 12 | 8 | 8 |
| Max Prompt Length | 200 | 512 | 200 | 200 |

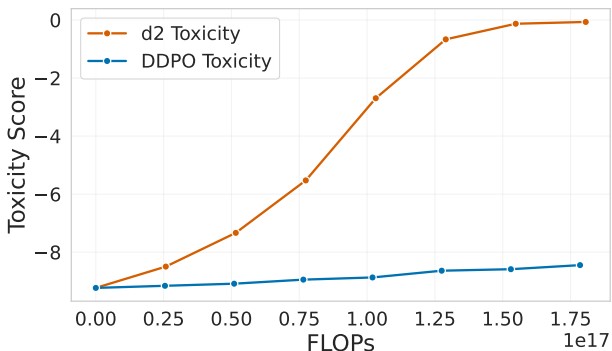

*Figure 10.* Toxicity steering results of d2-AnyOrder and DDPO on Eso-LM.

**FLOP Estimation.** We estimate computational consumption based on the operations executed during on-policy training loops. The total FLOP count for each loop consists of three components: on-policy sampling, likelihood evaluation for both the old and reference policies, and likelihood evaluation for the currect policy. We utilize the following estimation heuristics:

1. **On-policy Sampling:** $L \times G \times T \times 2P$, where $L$ is the sequence length, $G$ is the effective group size, i.e., group size times GPU numbers, $T$ is the number of sampling steps, and $P$ is the parameter count.

2. **Likelihood Evaluation for Old and Reference Policy:** $L \times G \times N \times 2P$ (per policy), where $N$ is the number of StepMerge segments.

3. **Likelihood Evaluation for Current Policy:** $L \times G \times N \times 4P$.

For sampling and likelihood evaluation, the multiplier is set to $2P$ per token, reflecting the cost of a forward pass with gradient computation disabled. For gradient updates, we adopt a multiplier of $4P$, accounting for the forward pass ($2P$) and activation backpropagation ($2P$) through the frozen backbone. We neglect the gradient overhead of the LoRA parameters, as they are negligible relative to the 8B parameters of the LLaDA backbone.

# D. Additional Experimental Results

### D.1. d2-AnyOrder

#### D.1.1. Eso-LM

**Main Result.** In Figure 10, we provide more data points collected in the Eso-LM toxicity steering experiment. We can see that the d2-AnyOrder curve strictly dominates the DDPO curve, demonstraing the efficacy of our proposed trajectory likelihood evaluation technique.

**Ablation.** In Figure 11, we provide more data points collected in the Eso-LM toxicity ablation experiment. We consistently observe a trend of d2-AnyOrder dominating d2-StepMerge.

#### D.1.2. Any-Order Causal LLaDA

**Finetune.** As shown in Table 10, applying the first finetuning stage on the Intellect-SFT data significantly increases the benchmark performance after the instruction-finetuning stage, demonstrating the advantage of increasing the diversity and

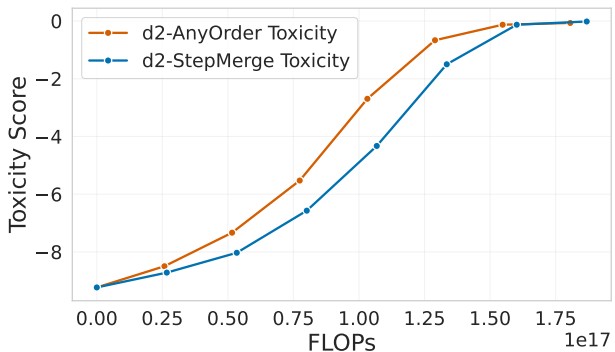

*Figure 11.* d2-AnyOrder v.s. d2-StepMerge on toxicity steering.

*Table 10.* Benchmark performance of any-order causal LLaDA with and without the Intellect-SFT finetuning stage.

| Finetuning Setting | GSM8K | MATH500 |
|---|---|---|
| GSM8K / MATH500 | 55.34% | 16.80% |
| Intellect-SFT + GSM8K / MATH500 | **59.21%** | **22.00%** |

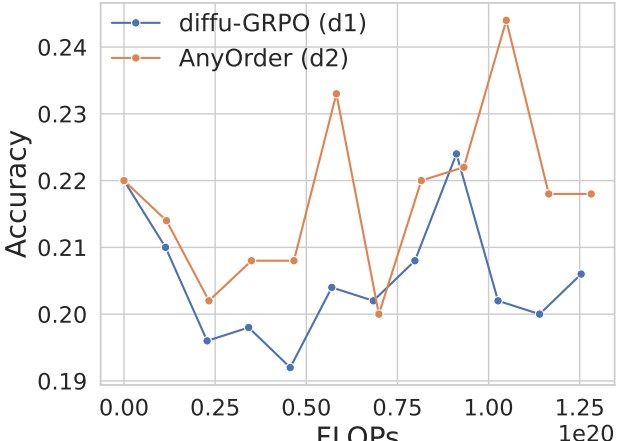

*Figure 12.* d2-AnyOrder v.s. diff-GRPO on the any-order causal LLaDA finetuned for MATH500.

scale of data that the model is exposed to during finetuning.

**RL.** In Figure 12, we present the performance-compute dynamics of d2-AnyOrder and diffu-GRPO on an any-order causal LLaDA checkpoint that we have on MATH500. d2-AnyOrder demonstrates a better trend to steer up the test set pass@1 than diffu-GRPO, further consolidating the efficacy of our proposed trajectory likelihood estimators.

## E. Assets

In Table 11, we list the datasets (and corresponding licenses, when available) used in this work. In Table 12, we list the software packages (and corresponding licenses) used in this work.

*Table 11.* Datasets (and corresponding licenses) used in this work.

| Dataset | License |
|---|---|
| OpenWebText (Gokaslan et al., 2019) | Creative Commons CC0 license ("no rights reserved") |
| GSM8K (Cobbe et al., 2021) | MIT |
| MATH500 (Lightman et al., 2024) | Apache-2.0 |
| INTELLECT-3-SFT (Ashz, 2024) | Apache-2.0 |

*Table 12.* Software (and corresponding license) used in this work.

| Library | License |
|---|---|
| HuggingFace (Wolf et al., 2019) | Apache 2.0 |
| Hydra (Yadan, 2019) | MIT |
| NumPy (Harris et al., 2020) | NumPy license |
| Matplotlib (Hunter, 2007) | Matplotib license |
| OmegaConf | BSD 3-Clause |
| Pandas (pandas development team, 2020) | BSD 3-Clause "New" or "Revised" |
| PyTorch (Paszke et al., 2019) | BSD-3 Clause |
| Seaborn (Waskom, 2021) | BSD 3-Clause "New" or "Revised" |
| TRL (von Werra et al., 2020) | Apache-2.0 |

