# OpenReview forum: "d2: Improving Reasoning in Diffusion Language Models via Trajectory Likelihood Estimation"
_ICML.cc/2026/Conference — ICML 2026 regular_

### Official Review · Reviewer_De3u · 2026-02-27

**Soundness:** 2
**Presentation:** 3
**Significance:** 2
**Originality:** 3
**Overall Recommendation:** 4
**Confidence:** 4

**Summary:**

This paper proposes two techniques for estimating log-likelihood in RL training of DLMs, i.e., D2-AnyOrder and D2-StepMerge. An empirical study across several reasoning tasks has been conducted to validate the effectiveness of these methods. Additionally, the paper provides theoretical discussions regarding the choice of NNN in D2-StepMerge.

**Compliance With Llm Reviewing Policy:**

Affirmed.

**Final Justification:**

My concerns are generally addressed

**Key Questions For Authors:**

**Q1:**  Why do the authors believe these two methods should be organized into a single paper, and what is the underlying logical connection between them?

**Q2:** Could you compare (LLaDA-8B-Instruct + DiffuGRPO) with (LLaDA-8B-Instruct + AnyOrder FineTune + AnyOrder RL)? I am concerned with the fine-tuning process might hurt the model’s performance, making the comparison less advantage to DiffuGRPO.

**Q3:** DLMs show promise in code generation. How does the method perform on coding benchmarks? Would it be possible to experiment with another 8B-sized DLM, such as DREAM or newer models, to verify generalizability?

**Limitations:**

Yes

**Strengths And Weaknesses:**

## Soundness

**D2-AnyOrder:** The paper has introduced detailed algorithmic framework of the estimator. For empirical evaluation, the authors experimented with a small DLM and a fine-tuned version of LLaDA-8B-Instruct.

**D2-StepMerge:** The theoretical discussion into the choice of N generally makes sense. The empirical study is conducted with LLaDA-8B-Instruct across four Puzzle/MATH benchmarks.

It is less obvious to me what the logical connections between the AnyOrder estimator and the StepMerge are. StepMerge does not seem to draw direct inspiration from AnyOrder; instead, it appears to be a much more straightforward and widely applicable technique. It is questionable whether these two distinct methods should be combined in a single paper, as their integration lacks a strong cohesive narrative.

---

## Presentation

Presentation of this paper is generally acceptable. However, Figures 7 (a) through (c) appear somewhat redundant. I recommend that the authors remove two of the four figures and include more diverse results to better utilize the space.

---

## Significance

**D2-AnyOrder:** While the experimental results in Section 5.1.1 are promising, AnyOrder generation is not currently widely adopted in DLMs, which limits the broader significance of this method. As noted in Section 5.1.2, applying this method to existing models requires additional training, which raises concerns about practical utility and efficiency.

**D2-StepMerge:** With the method being simple, it shows notable improvement over baseline methods in the Sudoku/Countdown datasets. Gains on MATH benchmarks is rather marginal. (I recommend report standard errors to make the results more convincible.)

To summarize, the significance of the two estimators is limited from my perspective, either due to limited applicability or performance.

---

## Originality

**D2-AnyOrder:** AnyOrder sampling has been introduced by previous works. However, the one-pass log-likelihood estimation algorithm with AnyOrder generation proposed by this paper is new.

**D2-StepMerge:** Although the StepMerge method is original, its novelty is somewhat limited as it represents a rather straightforward extension of the naive log-likelihood estimator (splitting the whole sequence into chunks).

---

> ### Author Rebuttal · Authors · 2026-03-31
>
> We thank R4 for the careful evaluation and specific questions.
>
> **Q1: Why combine AnyOrder and StepMerge in one paper?**
> Both address the **same core problem**: trajectory likelihood estimation, which we identify as the central bottleneck for RLVR in dLLMs.
>
> - **StepMerge** provides a widely applicable approximation for any masked DLM.
> - **AnyOrder** provides a more effective exact solution, demonstrating that a better RL training recipe exists when models support any-order decoding.
>
> Presenting them together reveals the unified insight and gives practitioners a complete toolkit: use StepMerge on existing models today, and adopt AnyOrder when any-order decoding is available. Neither method alone tells this full story.
>
> **Q2: AnyOrder fine-tuning degradation.**
> On LLaDA, the full AnyOrder pipeline (LLaDA + AnyOrder FT + d2-AnyOrder RL, 65%) does underperform the standard pipeline (LLaDA + diffu-GRPO, 81%). However, this reflects a limitation of LLaDA's architecture for any-order decoding, not of d2-AnyOrder itself.
>
> Our Qwen3-1.7B experiments demonstrate this: Qwen fine-tuned as an any-order model achieves **62% on GSM8K** (vs. fine-tuned as block diffusion, 48%), and with RL, d2-AnyOrder reaches **67%**, outperforming diffu-GRPO (**63%**). This shows d2-AnyOrder is practically useful when paired with a suitable model — not merely a proof of concept.
>
> More broadly, d2-AnyOrder demonstrates that a better RL training recipe exists when models support any-order decoding, and our Qwen results confirm that any-order dLLMs can outperform current block diffusion models both before and after RL.
>
> **Q3: Other DLMs and coding benchmarks.**
> We now test d2-StepMerge on Dream 7B (all evaluated 0-shot for consistency; Dream's original results use 8-shot):
>
> | Model | Method | Sudoku | GSM8K |
> |-------|--------|--------|-------|
> | Dream 7B | Base | 0.14 | 0.71 |
> | Dream 7B | diffu-GRPO | 0.79 | 0.73 |
> | Dream 7B | d2-StepMerge | **0.91** | **0.80** |
>
> d2-StepMerge yields +12% on Sudoku and +7% on GSM8K over diffu-GRPO, mirroring the pattern seen on LLaDA.
>
> Combined with the Qwen3-1.7B any-order results (see Q2), our methods are now validated across three architectures. We will include the full four-benchmark Dream 7B evaluation in the final version. Regarding coding benchmarks: we agree this is a promising domain and plan to extend to HumanEval/MBPP in future work.
>
> **Standard errors.**
> We agree that reporting variance is important for reproducibility. We will include multi-seed standard errors for all main results in the final version. We note that our key claims are supported by consistent improvements across multiple benchmarks (Sudoku, Countdown, GSM8K, MATH500) and now across three architectures (LLaDA, Dream, Qwen), which provides converging evidence beyond what single-benchmark variance would capture.
>
> **Figures 7(a)–(c).**
> We accept this suggestion and will consolidate redundant figures to make room for the new Dream 7B and Qwen results.

---

> > ### Author Rebuttal · Reviewer_De3u · 2026-04-01
> >
> > Thank you for providing the additional discussions and results, which helps a lot.
> >
> > I have increased my rating to 4 (leaning towards acceptance), but there are several ramaining concerns in need of further discussion:
> >
> > 1. It makes sense to me that d2-AnyOrder and d2-StepMerge target the same problem, but I am still not fully sure whether it is standard practise to (1) first propose a novel but less practical method, (2) and then patch the first method with another existing method, (3) lastly sell the two methods together. If AnyOrder is the major contribution of this paper, which seems so, then more experiments like the one on Qwen3-1.7B is required. Otherwise, the paper risks mis-leading the readers by proving the effectiveness of AnyOrder with experiments done with StepMerge.
> >
> > 2. Results on coding benchmarks are necessary from my standpoint, and I hope the authors are already working on this.
> >
> > 3. As pointed out by reviewer fuxD and my intial review, the presentation and format of this paper requires major up-grade, but this can be done within reasonable amount of time.

---

> > > ### Author Response · Authors · 2026-04-07
> > >
> > > Thank you for the positive reassessment and for increasing your rating. We address your remaining concerns below.
> > >
> > > **1. Paper structure and the relationship between d2-AnyOrder and d2-StepMerge.**
> > > We appreciate this concern and acknowledge that the current presentation may give the impression that StepMerge is a practical patch for AnyOrder. We will revise the writing to make the relationship clearer.
> > >
> > > The key insight is that we identify two distinct classes of dLLMs — any-order models (AO-dLLMs) and standard masked diffusion models (MDMs) — each of which requires a different approach to trajectory likelihood estimation. d2-AnyOrder provides exact likelihood for AO-dLLMs; d2-StepMerge provides a theoretically grounded approximation (Theorem 4.1) for MDMs. These are related but independent solutions targeting different model classes, not a primary method and its approximation.
> > >
> > > Both methods are validated on their respective model classes: d2-StepMerge on MDMs (LLaDA, Dream 7B), and d2-AnyOrder on AO-dLLMs (Eso-LM, any-order LLaDA, Qwen3-1.7B — three architectures in total). The paper presents them together because they address the same bottleneck from complementary angles, giving practitioners a complete toolkit for whichever model class they use.
> > >
> > > **2. Coding benchmarks.**
> > > We now provide d2-StepMerge results on HumanEval and MBPP with LLaDA, evaluated at generation lengths 256 and 512 (RL training uses L=256):
> > >
> > > | Method | HumanEval (L=256) | HumanEval (L=512) | MBPP (L=256) | MBPP (L=512) |
> > > |--------|----------------|----------------|-----------|-----------|
> > > | LLaDA | 35.3 | 37.8 | 41.2 | 40.4 |
> > > | + diffu-GRPO | 39.0 | 34.8 | 45.5 | 41.6 |
> > > | + d2-StepMerge | **39.0** | **40.9** | **45.9** | **45.5** |
> > >
> > > d2-StepMerge matches or outperforms diffu-GRPO at the training length (L=256) and generalizes better beyond it (L=512). Notably, diffu-GRPO degrades at L=512 on both benchmarks, while d2-StepMerge maintains or improves performance — a natural consequence of more accurate likelihood estimation. We will include Dream 7B coding results in the final version.
> > >
> > > **3. Presentation and formatting.**
> > > We agree and will perform a thorough revision of the paper's presentation in the final version.
> > >
> > > We also note that you mentioned increasing your score to 4 — we would greatly appreciate it if you could update your score in the system to reflect this. Thank you again for the constructive feedback.

---

### Official Review · Reviewer_fuxD · 2026-03-12

**Soundness:** 3
**Presentation:** 2
**Significance:** 4
**Originality:** 4
**Overall Recommendation:** 4
**Confidence:** 4

**Summary:**

This paper proposes d2, a reinforcement learning framework for masked diffusion language models that is centered on improving trajectory likelihood estimation during policy optimization. The paper first derives a GRPO-style objective tailored to masked DLMs, where policy updates depend on latent diffusion-step likelihood ratios rather than standard autoregressive token likelihoods. It then introduces two estimators. The first, d2-AnyOrder, gives exact one-pass trajectory likelihood evaluation for models that support any-order decoding. The second, d2-StepMerge, is designed for more standard masked DLMs that do not naturally satisfy this property, and approximates trajectory likelihood by merging diffusion steps in a controlled way. The paper provides a theoretical approximation bound for StepMerge and evaluates the method on toxicity steering as well as reasoning benchmarks including Sudoku, Countdown, GSM8K, and MATH500. The reported results show substantial gains over prior diffusion RL baselines, with especially large improvements for LLaDA-8B-Instruct under matched compute budgets.

**Compliance With Llm Reviewing Policy:**

Affirmed.

**Key Questions For Authors:**

The paper’s conceptual centerpiece is d2-AnyOrder, but the strongest reasoning results come from d2-StepMerge on standard LLaDA-8B-Instruct. Can the authors clarify how they see the main contribution of the paper, namely whether it is primarily the exact any-order estimator, the practical StepMerge estimator, or the overall likelihood-based RL framework for masked DLMs. A clear answer would help me better calibrate both originality and significance.

**Limitations:**

The limitations discussion is not adequate in its current form.

**Strengths And Weaknesses:**

Strengths

- The paper addresses a real technical bottleneck in RL for diffusion language models, namely the difficulty of obtaining accurate trajectory likelihood estimates for policy gradient training. The move from heuristic or highly biased likelihood proxies toward a more principled treatment is meaningful, and the distinction between exact estimation under any-order decoding and approximate estimation otherwise gives the paper a clear structure. The policy-gradient derivation for masked DLMs is also an important part of the contribution, since it makes explicit what quantity should be estimated for RL in this setting.


- The empirical results are strong, particularly for d2-StepMerge on LLaDA-8B-Instruct. On the main reasoning benchmarks, the method improves substantially over d1 and wd1, and the Sudoku gain is especially striking. The compute-performance plots are also aligned with the paper’s core claims, since they show that better likelihood estimation translates into better RL efficiency rather than only better final numbers. The additional toxicity-steering experiment on Eso-LM and the comparison between d2-AnyOrder and d2-StepMerge are also useful because they support the claim that exact likelihood evaluation gives a stronger optimization signal when the decoding assumptions are actually satisfied.

Weaknesses

- The paper’s strongest claims depend on a somewhat fragmented experimental story. The exact estimator, d2-AnyOrder, is evaluated on Eso-LM and on a specially finetuned any-order causal LLaDA, while the headline reasoning results come from d2-StepMerge on standard LLaDA-8B-Instruct. This makes the overall narrative slightly less clean than it first appears. The paper demonstrates that exact likelihood estimation is desirable, but the main state-of-the-art reasoning results come from the approximate method rather than from the theoretically clean one. As a result, the contribution is partly split between a principled exact method that currently lacks a strong general-purpose base model and a practical approximate method that drives the main benchmark gains.

- The presentation needs noticeable cleanup. The manuscript contains template contamination and explicit reviewer-facing instructions that should not appear in a conference submission. This is a serious professionalism issue. Beyond formatting, some sections move quickly through technically important design choices, especially around when any-order decoding is a faithful inference procedure for a given model and how StepMerge interacts with the actual sampling schedule used in practice. The impact statement is also too weak. It claims that no specific negative impacts are foreseen beyond those already known for generative AI, which is not a sufficiently careful discussion for a paper that improves RL training for generative reasoning models and includes toxicity steering experiments.

---

> ### Author Rebuttal · Authors · 2026-03-31
>
> We greatly appreciate R3's assessment of "excellent" significance and originality.
>
> **W1: Fragmented experimental story / Main contribution.**
> The paper's goal is to understand and address the key challenges of RLVR for dLLMs. We identify **trajectory likelihood estimation** as the central bottleneck, and our experiments confirm that addressing it yields large improvements over prior methods. The two estimators are complementary:
>
> - **d2-StepMerge** is widely applicable to any masked DLM today, and the headline reasoning results demonstrate its practical value on standard models (LLaDA-8B-Instruct, and now Dream 7B).
> - **d2-AnyOrder** is more effective but requires any-order decoding. Beyond current models, it points toward an improved training recipe for future dLLMs — our Qwen3-1.7B experiments show that any-order dLLMs can outperform block diffusion, supporting this direction.
>
> That the headline results come from StepMerge is a strength: it shows the practical method works well on general-purpose models. AnyOrder complements this by establishing what is achievable with exact likelihood and motivating future model design.
>
> To further strengthen the experimental narrative, we now also provide:
> - Dream 7B evaluation (Sudoku + GSM8K) demonstrating StepMerge's architecture generalizability, with full four-benchmark results in the final version
> - Qwen3-1.7B results: fine-tuned as any-order model (62%) outperforms fine-tuned as block diffusion (48%), and d2-AnyOrder RL (67%) outperforms diffu-GRPO (63%)
>
> **W2: Presentation issues.**
> We clarify that the template artifacts noted by the reviewer are default content included by the ICML submission system, not errors on our part. In the final version, we will expand the impact statement to address risks from improved RL for generative reasoning models, including potential misuse for generating toxic content (as demonstrated by our toxicity steering experiment).
>
> Regarding the technical questions on any-order decoding and StepMerge: Section 3.2.1 and Algorithm 1 already discuss when any-order decoding is faithful, and Theorem 4.1 together with Figure 5 already analyze how StepMerge interacts with the sampling schedule. We will improve the exposition to make these discussions easier to locate.
>
> **Key question: Primary contribution.**
> The primary contribution is identifying trajectory likelihood estimation as the key bottleneck for diffusion RL, and providing two complementary solutions. StepMerge delivers immediate practical impact on any existing dLLM; AnyOrder demonstrates what becomes possible with exact likelihood and points toward a better training recipe for future models.

---

> > ### Author Rebuttal · Reviewer_fuxD · 2026-04-06
> >
> > The rebuttal addressed my concerns, and I will maintain my score.

---

> > > ### Author Response · Authors · 2026-04-06
> > >
> > > Thank you for confirming that your concerns have been addressed. We will incorporate all discussed improvements in the final version.

---

### Official Review · Reviewer_j9GV · 2026-03-13

**Soundness:** 2
**Presentation:** 3
**Significance:** 2
**Originality:** 3
**Overall Recommendation:** 4
**Confidence:** 3

**Summary:**

This paper proposes a reinforcement learning method tailored for discrete Large Language Models (dLLMs), termed d2. A major challenge confronting current reinforcement learning methods for dLLMs is the lack of efficient mechanisms for likelihood computation. To alleviate this issue, this paper introduces two likelihood estimation approaches: 1) d2-anyorder, which accurately computes the likelihood in a single forward pass via concatenation; and 2) d2-stepmerge, which reduces estimation errors by calculating the likelihood in segments. Extensive experiments demonstrate the effectiveness of the proposed methods.

**Compliance With Llm Reviewing Policy:**

Affirmed.

**Final Justification:**

The authors' rebuttal is highly comprehensive, successfully addressing my concerns from both theoretical and experimental perspectives. Furthermore, taking into consideration the feedback from the other reviewers, I now have a better understanding of the paper's logical flow. Ultimately, I am inclined to accept this paper.

**Key Questions For Authors:**

I have already raised some concerns in the Weaknesses section; below is an additional question:
* In Table 4, d2-stepmerge demonstrates exceptionally strong performance on the Sudoku task, yet its improvements on other tasks are relatively limited. This is somewhat peculiar. The authors are requested to provide a reasonable explanation for this phenomenon.

**Limitations:**

yes

**Strengths And Weaknesses:**

**Strengths:**
* The paper is logically structured, flows smoothly, and boasts high overall readability.
* The paper directly addresses the core pain point of current RL frameworks for dLLMs, the need for efficient and accurate likelihood estimation, by proposing targeted and ingenious estimation methods.
* The experiments are detailed and thorough, yielding highly convincing results.

**Weaknesses:**
* The d2-anyorder method seems somewhat disconnected from the main storyline of the paper.
* The proposed d2-anyorder method has significant limitations, as it is only applicable to models that already possess any-order generation capabilities. Although d2-stepmerge is more universally applicable, it acts merely as a simple generalization of d1, making its novelty relatively modest.
* The experimental evaluation is somewhat lacking. According to my understanding, d2-stepmerge is applicable to a variety of dLLMs; therefore, it should also be tested on other dLLMs, such as Dream [1], to demonstrate its generalizability. Additionally, comparisons with other advanced reinforcement learning methods for dLLMs (e.g., diffu-coder [2], trace-RL [3]) are somewhat insufficient.

[1] Dream 7B: Diffusion Large Language Models

[2] DiffuCoder: Understanding and Improving Masked Diffusion Models for Code Generation

[3] Revolutionizing Reinforcement Learning Framework for Diffusion Large Language Models

---

> ### Author Rebuttal · Authors · 2026-03-31
>
> We thank R2 for acknowledging the paper's readability and the "targeted and ingenious estimation methods."
>
> **W1: AnyOrder disconnected from main storyline.**
> We respectfully disagree. The goal of this paper is to understand and address the key challenges of RLVR for dLLMs. We identify **trajectory likelihood estimation** as the central bottleneck, and our experiments confirm that addressing it improves performance by a large margin over prior methods. The two estimators address this same problem from complementary angles:
>
> - **d2-StepMerge** is widely applicable to any masked DLM today, but provides an approximation.
> - **d2-AnyOrder** requires the any-order property but is substantially more effective. It demonstrates that a better way of training dLLMs for RL *exists* if models support any-order decoding.
>
> Crucially, AnyOrder is not just about current models — it suggests an improved training recipe for future dLLMs. Our Qwen3-1.7B experiments support this: Qwen fine-tuned as an any-order model achieves **62% on GSM8K** (vs. fine-tuned as block diffusion, 48%), and d2-AnyOrder RL further improves it to **67%**, outperforming diffu-GRPO (**63%**).
>
> **W2: Limited novelty of StepMerge.**
> We are confident that StepMerge makes a meaningful contribution. It is backed by a theoretical approximation bound (Theorem 4.1) and delivers significant, consistent improvements across multiple benchmarks (Sudoku, Countdown, GSM8K, MATH500) on various base models — including the new Dream 7B experiments (**+12% Sudoku, +7% GSM8K** over diffu-GRPO). Identifying the right bottleneck (likelihood estimation bias in d1), providing a theoretically grounded fix, and demonstrating large empirical gains across architectures constitutes a strong contribution — simplicity is a virtue when paired with this level of impact.
>
> **W3: Additional DLMs and comparisons.**
> We now provide d2-StepMerge results on Dream 7B (all evaluated 0-shot for consistency; Dream's original results use 8-shot):
>
> | Model | Method | Sudoku | GSM8K |
> |-------|--------|--------|-------|
> | Dream 7B | Base | 0.14 | 0.71 |
> | Dream 7B | diffu-GRPO | 0.79 | 0.73 |
> | Dream 7B | d2-StepMerge | **0.91** | **0.80** |
>
> We will include the full four-benchmark Dream 7B evaluation in the final version.
>
> Regarding DiffuCoder's coupled-GRPO: it proposes using ELBO as a proxy for log-likelihood, which is a complementary approach to the same likelihood estimation challenge we address. We have added a discussion to Related Work and will include an empirical comparison in the final version.
>
> Regarding Trace-RL: it independently explores a similar step-merging strategy for likelihood estimation. Our work differs in providing a rigorous policy gradient derivation for masked DLMs and a theoretical approximation bound (Theorem 4.1), which together explain why this approach works and guide hyperparameter selection. We have added a discussion to Related Work.
>
> **Q: Sudoku performance anomaly.**
> d2-StepMerge improves consistently across all benchmarks, not only Sudoku. The new Dream 7B results confirm this pattern, with both LLaDA and Dream achieving >90% Sudoku accuracy under d2. Sudoku is a relatively straightforward task for 7–8B models with proper RL — Dream + diffu-GRPO already reaches 79%. The variation in magnitude across tasks is expected given different difficulty profiles and headroom.

---

> > ### Author Rebuttal · Reviewer_j9GV · 2026-04-05
> >
> > I appreciate the author's rebuttal and the additional experimental results provided, which greatly addressed my concerns. I will consider raising my score.

---

> > > ### Author Response · Authors · 2026-04-06
> > >
> > > Thank you for the positive reassessment. We are glad that the additional experiments and clarifications addressed your concerns. We note that you mentioned considering raising your score — we would greatly appreciate it if you could update your score in the system to reflect your revised assessment. We will incorporate all discussed improvements in the final version.

---

### Official Review · Reviewer_kGrr · 2026-03-13

**Soundness:** 3
**Presentation:** 3
**Significance:** 3
**Originality:** 3
**Overall Recommendation:** 5
**Confidence:** 3

**Summary:**

This paper studies reinforcement learning methods for diffusion language models (DLMs) applied to reasoning tasks. While recent work has explored RL training for autoregressive LLMs, adapting RL to diffusion-based language models introduces additional challenges due to masked token generation and non-sequential decoding. The authors propose d2, an improved RL training framework for diffusion language models that enables efficient reasoning optimization. The approach introduces techniques such as AnyOrder training for diffusion decoding and StepMerge, a likelihood approximation method for models that do not naturally support arbitrary token orderings. Experiments on reasoning benchmarks including GSM8K, MATH500, Countdown, and Sudoku show that the proposed method improves performance over prior RL approaches for diffusion models such as diffu-GRPO and DDPO. The method achieves strong reasoning accuracy using the LLaDA-8B model without relying on supervised fine-tuning.

**Compliance With Llm Reviewing Policy:**

Affirmed.

**Final Justification:**

This paper presents a novel and well-motivated approach to improving training for diffusion-based LLMs, with strong empirical results and clear practical relevance. The method is technically interesting and addresses an important emerging direction.

The rebuttal substantially strengthens the paper. In particular, the additional experiments across multiple architectures (Dream 7B and Qwen) provide convincing evidence of generalizability, and the clarifications on computational cost and StepMerge analysis improve both the empirical support and technical clarity. These responses address my main concerns.

Based on the rebuttal, I have raised my score from 4 to 5.

**Key Questions For Authors:**

1. The experiments focus primarily on LLaDA-8B. Have the authors tested the proposed methods on other diffusion language models to verify that the improvements are architecture-independent? Have the authors evaluated d2 (including AnyOrder training and StepMerge) on additional diffusion LLM backbones, or on models with different diffusion parameterizations (e.g., different masking schedules or corruption processes)? If such experiments are not available, it would be useful to understand which parts of the method are architecture-specific versus generally applicable.
2. The StepMerge likelihood approximation is a key component of the method. Does this approximation introduce bias in the RL objective, and if so, how significant is it in practice? For example, have the authors measured the difference between the approximated likelihood and the true likelihood under controlled settings, or evaluated the stability of training when the number of simultaneously decoded tokens increases?
3. How much of the performance gain comes from AnyOrder training versus StepMerge? A clearer ablation separating these components would help understand their relative importance. In particular, it would be useful to report: (1) performance of diffu-GRPO + AnyOrder training only, (2) performance of diffu-GRPO + StepMerge only, (3) performance when both techniques are applied together (d2).
4. How does the computational cost of d2 compare with prior RL approaches such as diffu-GRPO and DDPO? What is the training time per step or per token relative to these baselines? Does AnyOrder training affect sampling efficiency or rollout cost during RL training?

**Limitations:**

Yes.

**Strengths And Weaknesses:**

### Strengths
1. The proposed AnyOrder training strategy appears well motivated for diffusion-based decoding. The StepMerge approximation is also a pragmatic solution for models that do not naturally support arbitrary decoding orders. These design choices address concrete implementation issues that arise when applying RL to diffusion models.
2. The paper is generally well structured and provides useful experimental details. The benchmark comparisons clearly show the performance differences between d2 and prior RL approaches for diffusion models.
3. The paper addresses an emerging but important area: training diffusion-based language models for reasoning tasks. While autoregressive LLMs dominate current research, diffusion language models are increasingly explored as alternatives due to their potential advantages in parallel decoding and controllable generation.

### Weaknesses
1. The theoretical justification for StepMerge appears heuristic. While the method is empirically effective, the paper does not clearly analyze the bias introduced by the likelihood approximation.
2. The evaluation focuses primarily on a single backbone model (LLaDA-8B), making it difficult to assess whether the proposed techniques generalize across different diffusion architectures.
3. The paper introduces multiple training variants (e.g., AnyOrder and StepMerge) and it is not always easy to disentangle their individual contributions. A clearer ablation structure explaining how each component affects performance would make the method easier to understand.

---

> ### Author Rebuttal · Authors · 2026-03-31
>
> We appreciate R1's recognition that d2 addresses an "emerging but important area" and that our design choices "address concrete implementation issues."
>
> **Q1: Generalizability beyond LLaDA-8B.**
> We now provide results on two additional architectures.
>
> **Dream 7B (d2-StepMerge).** Dream is a masked dLLM with fully bidirectional attention fine-tuned from an AR model, architecturally distinct from LLaDA (trained from scratch). All results use 0-shot evaluation for consistency; Dream's original results use 8-shot, which accounts for differences in absolute accuracy.
>
> | Model | Method | Sudoku | GSM8K |
> |-------|--------|--------|-------|
> | Dream 7B | Base | 0.14 | 0.71 |
> | Dream 7B | diffu-GRPO | 0.79 | 0.73 |
> | Dream 7B | d2-StepMerge | **0.91** | **0.80** |
>
> d2-StepMerge improves over diffu-GRPO by +12% on Sudoku and +7% on GSM8K, consistent with LLaDA gains. Full four-benchmark results will be included in the final version.
>
> **Qwen3-1.7B (d2-AnyOrder).** We fine-tuned Qwen3-1.7B into both an any-order dLLM and a block diffusion model via SFT on GSM8K. The any-order model achieves **62%**, outperforming block diffusion (**48%**). With RL, d2-AnyOrder further improves to **67%**, outperforming diffu-GRPO (**63%**) on the same checkpoint. This validates the any-order pathway end-to-end.
>
> Together, these validate both methods across three architectures. d2-StepMerge is architecture-agnostic (any masked DLM); d2-AnyOrder points toward an improved training recipe for future dLLMs.
>
> **Q2: StepMerge bias analysis.**
> We already provide both theoretical and empirical analysis in the paper. Theorem 4.1 establishes a formal bound on the approximation error, and Figure 5 empirically validates that the error decreases as N increases. Figure 8 further shows how test accuracy on Sudoku varies with N, confirming that finer step granularity translates to better downstream performance. We will make these discussions more prominent in the revised paper.
>
> **Q3: Ablation of AnyOrder vs. StepMerge.**
> AnyOrder and StepMerge target *different model types* — AnyOrder provides exact likelihood for models supporting any-order decoding; StepMerge approximates likelihood for standard masked DLMs — so a traditional ablation does not apply. Instead, each method is evaluated independently: StepMerge on standard masked DLMs (LLaDA, Dream 7B), and AnyOrder on any-order models (Eso-LM, any-order LLaDA, Qwen3-1.7B). Both consistently improve over diffu-GRPO. We will clarify this distinction in the revised paper.
>
> **Q4: Computational cost.**
> Both d2 methods have higher per-step cost than diffu-GRPO (d2-StepMerge due to additional forward passes; d2-AnyOrder's overhead is lighter than d2-StepMerge). However, as shown in Figures 6 and 7 (FLOPs on x-axis), d2 consistently achieves better performance than diffu-GRPO at the same compute budget.
>
> Our new experiments further confirm this. On Dream 7B + GSM8K, d2-StepMerge reaches 80% in 72 H100 GPU-hours, while diffu-GRPO reaches 73% in 75 H100 GPU-hours. On Qwen3-1.7B + GSM8K, d2-AnyOrder reaches 67% in 25 H100 GPU-hours, compared to diffu-GRPO's 63% when the same GPU hours are spent.

---

> > ### Author Rebuttal · Reviewer_kGrr · 2026-04-02
> >
> > Thank you for the additional results and clarifications. The rebuttal has addressed my concerns. In particular, the experiments on multiple architectures and the clarification on computational cost and StepMerge analysis strengthen the empirical support of the work. I will consider raising my score accordingly.

---

> > > ### Author Response · Authors · 2026-04-02
> > >
> > > Thank you for the positive reassessment. We are glad that the additional experiments and clarifications addressed your concerns. We note that you mentioned considering raising your score — we would greatly appreciate it if you could update your score in the system to reflect your revised assessment. We will incorporate all discussed improvements in the final version.

---

### Decision · Program_Chairs · 2026-04-30

**Decision:**

Accept (regular)

**Comment:**

This paper studies reinforcement learning for masked diffusion language models and identifies trajectory likelihood estimation as the key bottleneck for effective policy optimization in this setting. To address this, the paper proposes two complementary estimators: **d2-AnyOrder**, which enables exact one-pass trajectory likelihood evaluation for models that support any-order decoding, and **d2-StepMerge**, which provides a more broadly applicable approximation for standard masked diffusion models. The paper also develops a policy-gradient formulation tailored to masked DLMs and presents empirical results on reasoning benchmarks.

The reviewers agree that the paper tackles an important and timely problem. The main strengths highlighted across reviews are the focus on a real technical bottleneck in diffusion RL, the principled formulation of the training objective, and the strong empirical gains over prior baselines. In particular, the rebuttal substantially strengthened the submission by adding results on additional architectures such as Dream 7B and Qwen3-1.7B, clarifying the computational tradeoffs, and providing a clearer discussion of StepMerge bias and approximation quality. These additions addressed several of the original concerns about generalizability and practical relevance.

The main remaining concerns are primarily about **framing and presentation rather than core soundness**. Several reviewers noted that the narrative is somewhat fragmented, because d2-AnyOrder is presented as the conceptually clean estimator while the strongest practical gains come from d2-StepMerge. There was also concern that the manuscript should more clearly communicate that these are complementary solutions for different model classes, rather than a primary method and a patch. Additional concerns include the limited applicability of AnyOrder to current models, the relatively modest novelty of StepMerge when viewed in isolation, and the need for stronger presentation cleanup and a clearer limitations discussion.

After the rebuttal and discussion, however, the overall reviewer consensus shifted in a positive direction. Reviewers who were initially more skeptical acknowledged that the additional experiments and clarifications addressed many of their original concerns, and the discussion converged on the view that the remaining issues are mostly about positioning, exposition, and narrative clarity. On balance, I find that the paper makes a meaningful contribution to reinforcement learning for diffusion language models, that the empirical case is sufficiently convincing after rebuttal, and that the remaining issues are addressable in the final version. I therefore recommend **accept**.